



# ITALICA, an extensive and accurate spatio-temporal catalogue of rainfall-induced landslides in Italy

Silvia Peruccacci[1], Stefano Luigi Gariano[1], Massimo Melillo[1], Monica Solimano[2], Fausto Guzzetti[1,3], Maria Teresa Brunetti[1]

[1]Istituto di Ricerca per la Protezione Idrogeologica, Consiglio Nazionale delle Ricerche, Perugia, 06128, Italy
[2]Agenzia Regionale per la Protezione dell'Ambiente Ligure, Genova, 16149, Italy
[3]Presidenza del Consiglio dei Ministri, Dipartimento della Protezione Civile, Roma, 00189, Italy

*Correspondence to*: Silvia Peruccacci (Silvia.Peruccacci@irpi.cnr.it)

**Abstract.** Italy is frequently hit and damaged by landslides, resulting in substantial and widespread disruptions. In particular,
slope failures have a high impact on the population, communication infrastructure and the economic and productive sectors. The hazard posed by landslides requires adequate responses for landslide risk mitigation, with special attention to the risk to the population. In 2006 the Italian Department of Civil Protection, an Office of the Prime Minister, commissioned the Research Institute for Geo-Hydrological Protection (*Istituto di Ricerca per la Protezione Idrogeologica*), a research institute of the Italian National Research Council, to carry out operational forecasting of rainfall induced-landslides.

Collecting landslide information in a catalogue is a preliminary action toward landslide forecasting. The use of spatially and temporally inaccurate landslide catalogues results in an uncertain and unreliable operational landslide forecasting. Consequently, accurate catalogues are needed to reduce the uncertainties, which are to some extent unavoidable. To this end, over the last 15 years many researchers have been involved in compiling a catalogue called ITALICA (ITAlian rainfall-induced LandslIdes CAtalogue), which currently lists 6312 records with information on rainfall-induced landslides that occurred over

the Italian territory between January 1996 and December 2021. Overall, more than a third of the catalogue has very high geographic accuracy (less than 1 km$^2$) and hourly temporal resolution. In contrast, less than 2% of the catalogue have low and very low geographical accuracy and daily temporal resolution. This makes ITALICA the largest catalogue of rainfall-induced landslides accurately located in space and time available in Italy. Without this high level of accuracy, the precipitation responsible for the initiation of landslides cannot be reliably reconstructed, thus making the prediction of landslide occurrence

ineffective.

ITALICA's information on rainfall-induced landslides in Italy places a special emphasis on their spatial and temporal location, making the catalogue especially suitable for defining the rainfall conditions capable of triggering future landslides on the Italian territory. This information is fundamental for decision-making in landslide risk management.



## 1 Introduction

Italy has a long history of landslides and of related catastrophes. Landslides are complex and diverse phenomena triggered by multiple causes, including natural (meteorological or geophysical) and anthropogenic ones. Rainfall-induced landslides are more widespread than any other geological event, and occur anywhere in Italy with serious consequences for people and property. Between 1972-2021, landslides caused 145,548 homeless and evacuees and 2504 casualties (Polaris report, Bianchi and Salvati, 2023, https://polaris.irpi.cnr.it/report/last-report/, last access 2 February 2023, in Italian). In the four-year period

2017-2020, the Italian Institute for Environmental Protection and Research (ISPRA) counted 645 "major landslide events", defined as those that cause deaths, injuries, evacuations, and damage to buildings, cultural heritage, and infrastructures (Trigila et al., 2021).

To mitigate landslide risk in Italy, in 2006 the Italian Department of Civil Protection (DPC), an Office of the Prime Minister, commissioned the Research Institute for Geo-Hydrological Protection of the Italian National Research Council (CNR IRPI) to

carry out operational forecasting of rainfall induced-landslides.

The prediction of the possible spatial and temporal occurrence of shallow rainfall-induced landslides in large areas is accomplished by using empirical rainfall thresholds (Guzzetti et al., 2022). This simplified approach aims to identify an empirical relationship between rainfall and the occurrence of landslides, explicitly neglecting knowledge of the physical laws governing slope instability mechanisms. Thresholds are calculated on a statistical basis by compiling historical catalogues of

past documented failures and analysing the triggering rainfall conditions. They are well suited for predicting the occurrence of shallow landslides, for which the correlation between rainfall and landslide initiation is direct. Thresholds are not as effective in predicting the occurrence of deep-seated landslides because of the lack of specific information and knowledge on the behaviour and hydrological characteristics of the subsurface of unstable slopes.

Although statistical and probabilistic methods for defining reliable and reproducible empirical rainfall thresholds are well-

established (e.g., Berti et al., 2012; Segoni et al., 2014; Melillo et al., 2018), the availability of information required for the development of sub-regional and local regional thresholds is not as satisfactory. More effort and resources need to be devoted to the retrieval of information on rainfall events that have triggered landslides. To this end, since 2007 CNR IRPI has been involved in collecting historical data of rainfall-induced landslides in order to identify the critical triggering conditions. This activity has involved many of the institute's researchers over the years in different geographical and climatic contexts of Italy.

The use of a standardised methodology for collecting and classifying data has resulted in a homogenous catalogue that includes accurate information on the geographic location and timing of landslide initiation. The catalogue is called ITALICA (ITAlian rainfall-induced LandslIdes CAtalogue) and currently lists 6312 records with information on rainfall-induced landslides that occurred over the Italian territory between January 1996 and December 2021.





## 2 Background

Numerous examples of national landslide catalogues, databases, or inventories are available in the literature, a brief review of
which is given in this work. The Oxford Learner's English Dictionary defines a "catalogue" as "a long series of things that
happen (usually bad things)", a "database" as "an organised set of data that is stored in a computer and can be looked at and
used in various ways", an "inventory" as "a written list of all the objects, furniture". In particular, a landslide inventory is
defined as "a record of recognized landslides, distinguished by typology, geometry and activity, in a particular area"
(Corominas et al., 2015).

A bibliographical and archive inventory of landslides and floods in Italy (southern Europe) covering the period 1917–1990
was prepared as part of the AVI – *Aree Vulnerate Italiane* (an acronym for Areas Affected by Landslides or Floods) national
project (Guzzetti et al., 1994); subsequently the inventory was upgraded to cover the period 1900–2002 (Guzzetti and Tonelli,
2004). (Guzzetti, 2000) compiled a catalogue of historical landslides with consequences for the Italian population from 1279
to 1999. The catalogue was revised and expanded by (Salvati et al., 2010, 2018). A digital landslide database was prepared for
Nicaragua (central America), containing spatial information for approximately 17,000 landslides that occurred in the period
1826–2003 (Devoli et al., 2007). Information was searched from historical documents, technical reports and inventory maps,
and included: date; location; landslide type; trigger; meteorological, geological, and morphological details; and damage. The
IFFI Project (an Italian acronym for the Inventory of Landslide Phenomena in Italy) was launched in 1999 with the aim of
identifying and mapping landslides over the national territory (Trigila et al., 2010). As of 2022, the IFFI inventory contains
620,808 landslides, covering an area of approximately 23,700 km$^2$, about 8% of the Italian territory. Van Den Eeckhaut and
Hervás (2012) published a detailed analysis of national landslide databases existing (at the time) in Europe. They found that
22 out of the 37 European countries contacted had national databases, containing a total of 633,696 landslides, of which
485,004 were located in Italy. Most of these databases were geomorphological inventories, which therefore did not contain
temporal information on landslide occurrence and details about the triggers. Information on landslide locations was collected
by traditional methods such as field surveys, interpretation of aerial photos and analysis of historical documents. A landslide
database for Great Britain (western Europe) was developed by the British Geological Survey (Foster et al., 2012; Pennington
et al., 2015), relying upon a variety of sources including maps, other databases, reports, research theses, and newspaper articles.
It included over 17,000 records of landslide events with more than 35 attributes, comprising location, landslide size and type,
trigger mechanism, damage, material, and occurrence date. Mrozek et al. (2014) published an inventory of landslides in Poland
(central Europe), containing about 40,000 landslides, covering 1031.9 km$^2$, mapped in 161 municipalities, until April 2014.
The inventory contained spatial information (mostly collected during geomorphological field mapping) on landslide location
and size, as well as information on triggering events and landslide-related damage. Temporal information on the phenomena
was lacking. A national landslide database for Germany (central Europe) was produced by Damm and Klose (2014, 2015) that
included 1,720 landslide events (more than 13,000 individual data files) caused by several triggers during the period 1820–
2013, and was based on different sources such as scientific publications, field data, and agency archives. The data collected in





their database included information on the occurrence date or time of the failures; geographic coordinates and landslide location; administrative region, as well as available data sources. Komac and Hribernik (2015) presented the national landslide inventory of Slovenia (central Europe), which at the time of publication contained a total of 6234 records entered into the

database in point format. The national database put together information from administrative sources such as the Slovenian Geological Survey, Environment Agency, Roads Agency, and several municipalities. Various details were included in the database, with emphasis on landslide sizes and volumes. Rosser et al. (2017) prepared a landslide database for New Zealand, bringing together existing landslide data stored in a variety of sources including aerial photographs, field and media reports, and proposing a unified data model. The database comprised 22,575 landslide records (mapped as either points, lines, or

polygons) including information on locations, timing, type, triggering event, volume and area data, and consequences when available. Innocenzi et al., (2017) compiled a database containing information on 1054 landslides that occurred in Italy in the 4-year period 2012–2015, by searching the internet using Google Alerts (https://www.google.com/alerts). Each landslide was assigned a location, a date (daily resolution), a region and a nearest city; only 808 landslides had geographic information. Consulting online news sources from 2010 onwards, Calvello and Pecoraro (2018) published a georeferenced catalogue of

8931 landslides affecting the Italian territory from 2010 to 2017. Information collected in the catalogue includes: location and occurrence day, source of information, and number of landslides in case of areal events. Indeed, the records were classified as "single landslide events" (records only reporting one landslide) and "areal landslide events" (records including multiple landslides triggered by the same cause in the same area). Events were also classified in three classes according to the damage caused (very severe, severe, or minor). The Swiss Federal Research Institute compiled a database of naturally triggered floods,

landslides, and debris flows with a particular focus on the financial damage caused by such events (Andres and Badoux, 2019). The national database is also based on comprehensive regional landslide inventories (Hess et al., 2014). At the time of publication, the database contained 3690 landslides and 660 debris flows occurred in Switzerland in the period 1972–2016. The minimum information stored in the database was: date, time, location, municipality and canton, trigger, number of dead, injured or evacuated people, and estimation of the caused damage. A historical landslide database for Czechia (central Europe)

was compiled by Bíl et al. (2021), counting 699 records over the period 1132–1989. The records were characterised by several attributes, among which the type, location, beginning and end of movement, accuracy and source were mandatory. Information was gathered from national and local chronicles, technical reports, and photo interpretation. A national landslide inventory for Denmark (northern Europe) was prepared by Luetzenburg et al. (2022) based on a manual expert-based mapping approach on a high-resolution DEM and orthophotos. Overall, the inventory contained 3202 landslide polygons with attributes regarding

location, size, type of movement, and accuracy. Information on the time of occurrence of the phenomena, as well as their triggering causes, were not systematically included.

Two main global catalogues were compiled and published. Kirschbaum et al., (2010) compiled a catalogue of global-scale rainfall-triggered landslides that occurred between 2003 and 2008, drawing on news, scientific articles, and related hazard databases. A methodology to catalogue landslide events was also presented. The catalogue was subsequently updated, reaching

5741 records in the period 2007–2013 (Kirschbaum et al., 2015). Froude and Petley, (2018) published the Global Fatal



Landslide Database, collecting 4862 non-seismic landslides that caused 55,997 deaths worldwide from January 2004 to December 2016. Information was collected mostly from mass media reports, and secondarily using government and aid agency reports, scientific articles. The records include the date of occurrence and the location (coordinates and Country) of the landslides; the number of fatalities and injuries, and the trigger. Looking at Europe, Haque et al., (2016) presented the European

landslides database containing 476 fatal landslides that affected 27 European countries from January 1995 to December 2014, resulting in 1370 deaths.

As for Italy, none of the available catalogues have a level of accuracy as high as ITALICA. This characteristic makes it particularly suitable for use in operational landslide forecasting at the regional scale.

## 3 Study area

Italy is a boot-shaped peninsula, which covers 301,336 km$^2$ in southern Europe, from 7° to 19° E, and from 37° to 47° N (Fig. 1). Physiographically, Italy is characterised by two main mountain chains, the Alps and the Apennines. The Alps sweep in a west-to-east arc covering the northern tip of the country and extend 1200 km from E to W reaching an altitude of over 4,800 m a.s.l. and dividing the Italian peninsula from the rest of Europe. The Apennines is a mountainous and hilly chain extending longitudinally from NW to SE for 1200 km along the Italian peninsula. Elsewhere Italy deeps into the Mediterranean Sea and,

in particular it is surrounded by the Adriatic, Ionian, Thyrrenian, and Ligurian Seas, which are home to numerous islands, the largest of which are Sicily and Sardinia.

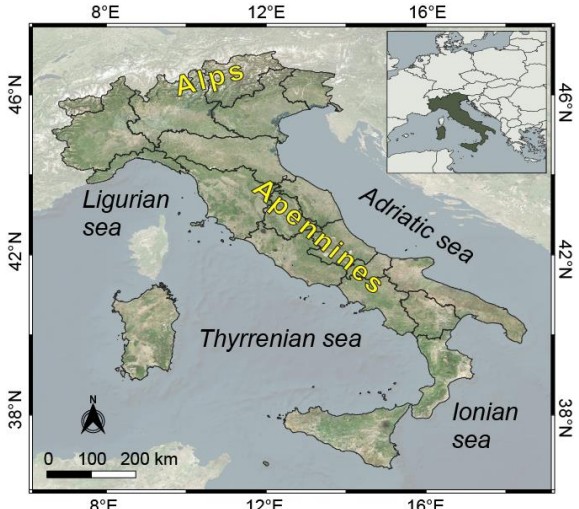

**Figure 1. Study area. Background from © Microsoft; EPSG: 4326.**

Italy is almost entirely seismically active, being located at the meeting point of the Eurasian and African plates. Sedimentary, metamorphic and igneous rocks of Paleozoic to Recent age are present, covered by different soil types with thicknesses from < 1 m to several metres. Given the conformation and moderate variation in latitude of the territory, the climate in Italy is quite





variable. In the North it is generally colder and wetter, and locally alpine in the mountainous area. Along the peninsula, the climate is temperate, with the duration of dry summers increasing toward the South. The eastern alpine and prealpine areas,

the northern Apennines have higher precipitation, with mean annual precipitation values exceeding 2,000 mm. In contrast, areas with lower precipitation, between 400 and 600 mm per year, are mainly found in southern Sicily, Puglia and southern Sardinia. Almost everywhere in Italy, November and July are the wettest and driest months, respectively (Fioravanti et al., 2022).

The abundance of relief and climatic characteristics make landslides a frequent and widespread phenomenon in Italy, where

they are triggered primarily by rainfall, secondarily by rapid snowmelt, and earthquakes (Guzzetti, 2000; Guzzetti and Tonelli, 2004).

## 4 Data and methods

To introduce the data used for the catalogue, it is important to state that the collected landslides are mostly those that had a direct or indirect impact on the population (structures and facilities, such as buildings, roads, railways). Landslides that

occurred in not-anthropized areas or for which there are no reports are seldom included in the catalogue.

Data on landslides are, in general, difficult to retrieve and not fully reliable in terms of completeness and temporal and spatial accuracy. In order to gather in-depth and up-to-date information on the location and timing of landslides in Italy, it has been necessary to concentrate efforts and human resources on various sources of information. Information was collected through the systematic reading of local newspapers, both printed and electronic, blogs and online information sources; consultation of

texts and periodicals held in municipal libraries and newspaper libraries; event reports, and reports following surveys; research in archives of different level of government (municipal, provincial and state); examination of online archival sources available from research organisations (e.g., SICI: an Italian acronym for Information System on Hydrological and Geomorphological Catastrophes; Guzzetti and Tonelli, 2004); consultation and involvement of Institutions in charge of land management, protection and surveillance (e.g., Regional Functional Centres, Provincial Commands of the National Fire and Rescue Service

and State Forestry Corps).

The analysis of sources on rainfall-induced landslides is challenging because the information is generally incomplete, uneven and sometimes conflicting. For example, one source allows the accurate location of a landslide to be pinpointed by showing one or more photos of the surrounding landscape, or of the kilometre marker in the case of a slope failure along a road. Another source may instead report details on the time or part of the day when the landslide occurred (late morning, evening); still others

may indicate the type of the landslide movement. Hopefully, with the help of the different sources, the reconstruction of where and when the landslide occurred can be established. In the event of a discrepancy, the search for additional sources continues until an agreed reconstruction is found. In the absence of minimum information such as the broad location and the day of occurrence of the landslide, the event is discarded. Landslides were excluded from the catalogue if: (1) the triggering factors were unknown or other than rainfall; (2) there was evidence of other causes operating along with rainfall in the activation (e.g.,



freeze-thaw cycles, rain-on-snow or snowmelt, seismic vibrations, anthropogenic influence); (3) the landslide location both in space or time had low accuracy, thus preventing the likely reconstruction of the triggering rainfall conditions (Palladino et al., 2018).

In the catalogue, for each record the information includes: (i) source of information; (ii) landslide type (if available from the source of information); (iii) landslide location (coordinates, municipality, province, region, geographic accuracy); (iv)
temporal information (day, month, year, time, date, temporal accuracy). Table 1 summarises the main fields included in ITALICA.





**Table 1. Summary of fields included in the ITALICA.**

| Category | Information on category |
|---|---|
| ID | Unique ID for each reported rainfall-induced landslide. |
| Information sources | Source of report information, including news reporting (NR) and institutions reporting (IR). |
| Landslide type | Landslide types are included if known or specified in the source and includes: debris flow (DF), earth flow (EF), mud flow (MF), rock fall (RF), generic shallow landslide (SL). |
| Longitude & latitude | Longitude and latitude of the reported failures. |
| Municipality, Province & Region | Municipality, Province and Region in which the landslide occurred. |
| Geographic accuracy | This field assigns a qualitative level for the landslide geographic accuracy based on the area over which the landslide realistically occurred, described as a radius from the coordinates of the failures (in kilometres):<br>• very high, $P_0$ exact landslide location;<br>• high, $P_1 < 1$ km$^2$;<br>• medium, $1 \leq P_{10} < 10$ km$^2$;<br>• low, $10 \leq P_{100} < 100$ km$^2$; and<br>• very low $100 \leq P_{300} < 300$ km$^2$. |
| Day, Month & Year | Reported day, month and year of the landslide, in separate columns. |
| Local Time | Reported hour and minute of the landslide, recorded as HH:MM (24 clock, local time). This field may also include an approximate time of day if known (e.g. morning, early/late morning, afternoon, evening, night, etc.). |
| Local date | This field summarises the date and local time of the reported landslides. |
| UTC date | This field summarises the date and the UTC time of the reported landslides. |
| Temporal accuracy | This field assigns a qualitative level for the landslide temporal accuracy, in three classes:<br>• level $T_1$ when the time (minute to hour) of the failure is known;<br>• level $T_2$ when the part of the day (e.g., early or late morning, midday, early or late afternoon, middle of the night) is known or the time is inferred from the online news publication;<br>• and level $T_3$ when only the day of occurrence is known. |

The information sources were classified into two categories: (i) news reporting, and (ii) Institutions reporting. News reporting includes information from online and printed newspapers, news sites, social media and blogs. Information from newspapers was initially gathered through systematic search of regional and local online archives. For this purpose, we used Google Alerts, which allows receiving alerts whenever a predefined keyword or combinations thereof is mentioned somewhere on the web. We used various terms linked to bad weather conditions with all possible synonyms of landslides (Table 2). The Google Alerts

search returns results when a rainfall-related term (e.g., "rainfall", "downpour") and a landslide-related term (e.g., "mass movement", "collapse") are found simultaneously in a webpage. Specifically, Table 2 shows all the possible 136 combinations searched for on the internet. The same search was done by including the plural of terms, if any.



**Table 2. Key terms used to select information on rainfall-induced landslides in Google Alerts search tool. The double and triple check marks indicate the number of possible cross-combinations.**

| | Rainfall | Cloudburst | Precipitation | Bad weather | Downpour | Shower | Flash flood | Storm |
|---|---|---|---|---|---|---|---|---|
| Landslide/slide | ✓✓ | ✓✓ | ✓✓ | ✓✓ | ✓✓ | ✓✓ | ✓✓ | ✓✓ |
| Landsliding | ✓ | ✓ | ✓ | ✓ | ✓ | ✓ | ✓ | ✓ |
| Mass movement | ✓ | ✓ | ✓ | ✓ | ✓ | ✓ | ✓ | ✓ |
| Slope failure/instability | ✓✓ | ✓✓ | ✓✓ | ✓✓ | ✓✓ | ✓✓ | ✓✓ | ✓✓ |
| Collapse | ✓ | ✓ | ✓ | ✓ | ✓ | ✓ | ✓ | ✓ |
| Boulder | ✓ | ✓ | ✓ | ✓ | ✓ | ✓ | ✓ | ✓ |
| Slump | ✓ | ✓ | ✓ | ✓ | ✓ | ✓ | ✓ | ✓ |
| Earth/debris/mud flow | ✓✓✓ | ✓✓✓ | ✓✓✓ | ✓✓✓ | ✓✓✓ | ✓✓✓ | ✓✓✓ | ✓✓✓ |
| Earth/Debris slide | ✓✓ | ✓✓ | ✓✓ | ✓✓ | ✓✓ | ✓✓ | ✓✓ | ✓✓ |
| Rock fall/slide/avalanche | ✓✓✓ | ✓✓✓ | ✓✓✓ | ✓✓✓ | ✓✓✓ | ✓✓✓ | ✓✓✓ | ✓✓✓ |

News about landslides was also retrieved from social media posts often accompanied by photographs taken shortly after the failure. Examples of such information are blogs, Facebook and Twitter posts from users who experienced traffic jams due to

landslides blocking roads. This kind of information makes it possible to better locate landslides in both space and time.

Institutions reporting comes from interventions following weather-related landslides carried out by institutional authorities, including the provincial Commands of the National Fire and Rescue Service, the regional civil protection functional centres and State Forestry Corps,  News about road disruptions caused by geo-hydrological phenomena were provided by ANAS (*Azienda Nazionale Autonoma delle Strade*), an Italian company that manages national road and motorway network, and

CCISS (*Centro Coordinamento Informazioni Sicurezza Stradale*), an Italian agency that provides traffic and travel information. Information about landslides occurring along the Italian railway network are provided by RFI (*Rete Ferroviaria Italiana*). Institutional authorities have proved particularly useful, as they provide reliable and accurate information about the exact or approximate landslide location in space and time, usually being the first responders at the scene of the event. Institutions reporting were often cross-referenced with news derived from chronicle sources, and this made it possible, in most cases, to

improve the temporal and spatial accuracy of the slope failures.

Where available, information on the landslide type was also collected. This was a critical task, because some of the sources (e.g., newspapers, firefighter reports, blogs) often used untechnical and therefore imprecise language to describe a slope failure. According to the categories defined by Cruden and Varnes (1996), we  classified the landslides as debris flow (DF),  earth-flow (EF), mud-flow (MF), rock fall (RF), and generic shallow landslide (SL) in the cases where the description of the type of

landslide was missing in the information sources.

The catalogue provides the geographic coordinates (longitude and latitude in WGS84) of individual failures along with administrative information, including municipality, province and region. Landslides then result as points on a map (not polygons) with an associated geographic accuracy depending on the type and quality of the information. Using the details provided in the information sources, the landslides were mainly located using Google Earth to retrieve their coordinates, taking

advantage of its multi-temporal set of images. The services of the Italian National Geoportal



(http://www.pcn.minambiente.it/viewer/), which allow all the maps (1:25,000 scale) provided by the Italian Army Geographical Support Office to be viewed, were also used to search for some ambiguous or unknown toponyms.

From Peruccacci et al. (2017), we identified five categories of decreasing geographic accuracy P: $P_0$ (very high); $P_1 < 1$ km$^2$ (high); $1 \leq P_{10} < 10$ km$^2$ (medium); $10 \leq P_{100} < 100$ km$^2$ (low); and $100 \leq P_{300} < 300$ km$^2$ (very low). The geographic accuracy
is assigned based on the maximum circular area within which the landslide realistically occurred. A level $P_0$, corresponding to the exact location of the landslide, was assigned to those failures for which the information source directly reported the geographic coordinates, or the road with an exact kilometric indication, or even, in the case of a landslide occurring in a built-up area, the street and the approximate house number. In particular, the road kilometre was obtained by searching for kilometre markers in Google Street View; in few cases, the landslide body was clearly sighted. Level $P_1$ corresponds to a landslide
located within a radius of less than about 0.6 km (i.e., 1 km$^2$ or less), e.g. the name of the street was known, but not the exact location. A medium level of geographic accuracy $P_{10}$ was assigned when the information obtained from the source allowed the identification of a large road sector, or a city block affected by the landslide (within a radius of less than about 1.8 km). A low level of geographic accuracy $P_{100}$ was attributed when the information source mentioned the district, borough or hamlet of a municipality where the landslide occurred.

The date of occurrence (year, month, and day) is given for each failure. As for the geographic accuracy, we defined a temporal accuracy T in three classes: $T_1$ when the time (from minutes to one hour) of the event is known; $T_2$ when the part of the day or the inferred time is known; $T_3$ when only the day of occurrence is known. The time of the $T_1$ class can be derived from both news reporting and institutional reporting, assuming that the authorities involved (e.g., Fire and Rescue Service, RFI) are warned immediately after the landslide event. The $T_2$ class is assigned in two cases. When the news specifies that the landslide
occurred in a time slot (e.g., late morning, early afternoon), an inferred time is given according to Table 3, which provides four main subdivisions of the day into nine time slots. In the case of online news reporting, the time at which the news was first published is used to determine the inferred time when the failure took place, assuming that the landslide certainly occurred before the news was posted. Lastly, where the news reports only the day on which the event occurred, the landslide is assigned a daily temporal accuracy $T_3$ and is conventionally assumed at the end of the day (23:59).


**Table 3. Inferred time of the landslide based on the time slot derived from the sources.**

| Time slot | Inferred time |
|---|---|
| Early morning | 8:00 |
| Morning | 11:00 |
| Late morning | 13:00 |
| Early afternoon | 15:00 |
| Afternoon | 17:00 |
| Late afternoon | 19:00 |
| Evening | 21:00 |
| Late evening | 23:59 |
| Night | 05:00 |



Information was collected and entered into the catalogue by several operators, who were assigned one or more administrative regions within which to conduct the search. The size of the team varied over time, with a minimum of five and a maximum of
nine operators working simultaneously. In order to limit subjectivity in the compilation of the catalogue, several workshop and training courses were organised to ensure the adoption of uniform criteria by all operators. A validation of the landslides added in the catalogue was carried out by assigning a random sample of the records from one operator to another one. In most cases the records were filled in with the same details. In case of discrepancies, they were double-checked by the team.

The catalogue records were stored in a spreadsheet and converted in comma-separated-values (.csv) and geoPackage (.gpkg)
files to be analysed and visualised in a GIS environment.

## 5 Description of the catalogue

ITALICA lists 6312 records with information on rainfall-induced landslides that occurred over the Italian territory between January 1996 and December 2021. Figure 2a shows the distribution of the 6312 slope failures classified by type (according to Cruden and Varnes, 1996). The landslides are fairly evenly distributed in the mountainous and hilly areas of the Country. Some
areas exhibit a higher concentration of events due to specific agreements with local authorities. Overall, about three-quarters (4762) of the catalogued mass movements were classified as generic shallow landslides (SL); while 13% (818) of the phenomena are rock falls (RF), which are homogeneously distributed over the whole territory. Debris, earth and mud flows (DF, EF and MF, respectively) cover together less than 12% (732) of the catalogue. DF are mainly located in the northern part of the Country, particularly in the Alps mountain chain. Figure 2b shows the number of landslides collected in each of the 20
administrative Italian regions. Overall, half of the regions count more than 200 landslides. In two regions, namely Liguria and Marche, more than 1000 landslides were collected, thanks to specific agreements with the regional civil protection offices.

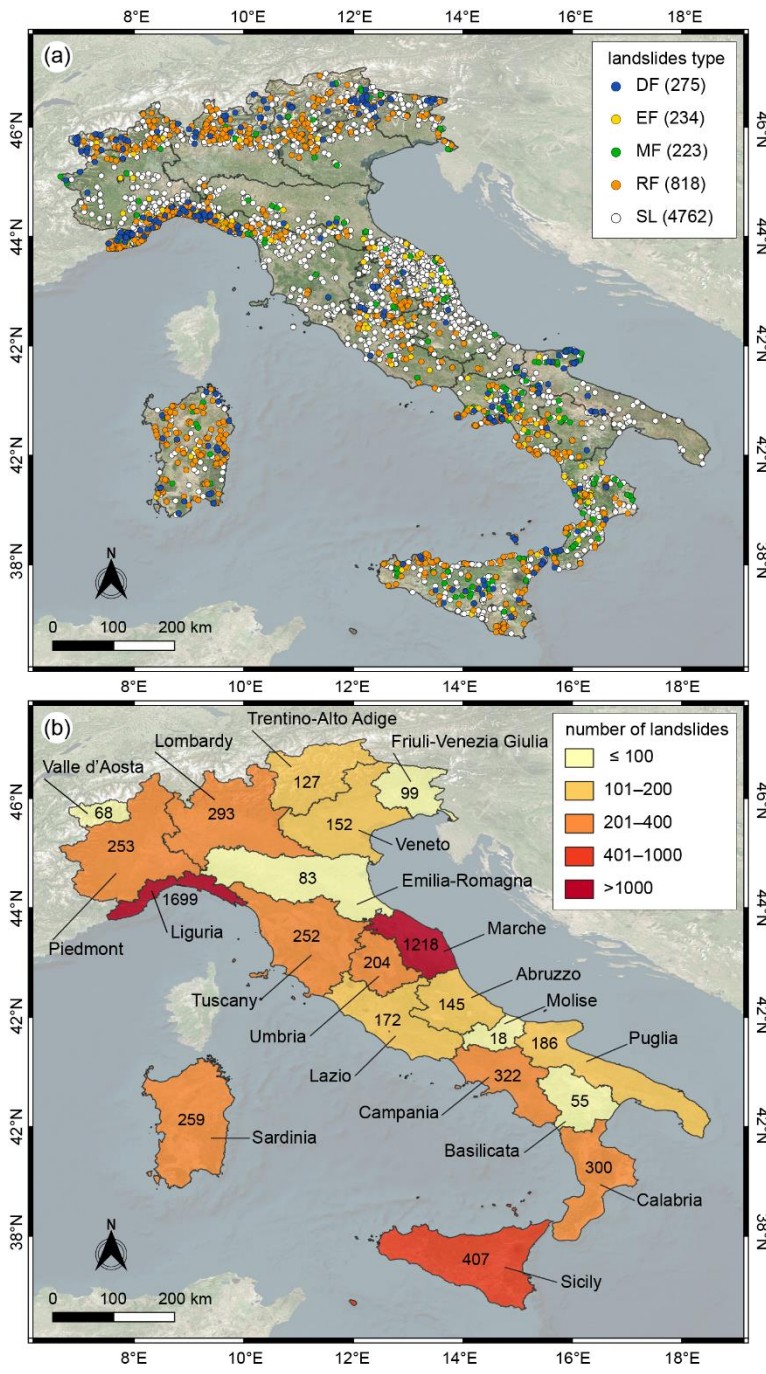

**Figure 2. (a) Location of the catalogued landslides, classified according to type. Key: DF, debris flow; EF, earth flow; MD, mud flow; RF, rock fall; SL, unspecified shallow landslide. The number of landslides for each type is given in brackets. (b) Number of landslides for each of the 20 Italian administrative regions (names in Italian). Colours of the regions are associated with the number of landslides in 5 classes. Background from © Microsoft; EPSG: 4326.**

Figure 3 shows the monthly distribution of the landslides, grouped by season. November, May, December, March, and January
are in descending order characterised by high variability in the number of landslides. Table 4 lists the mean, median and total
number of landslides per month. November is the month with the highest statistics on the number of failures. The monthly
median varies by a factor 2.5, thus evidencing the seasonality of the process. The difference between the mean and the median
values is significant for all months, indicating that the monthly distributions are not normal, as depicted by the violin plots in
Figure 3. Summer months (JJA) are characterised by a lower variability. Overall, 35.2% of the catalogued landslides occurred
in autumn (SON), 27.5% in winter (DJF), 23.9% in spring (MAM), and only 13.4% in JJA. Landslides that occurred in JJA
are mostly located in the Alps while those that occurred in DJF are mainly found in the Apennine chain.

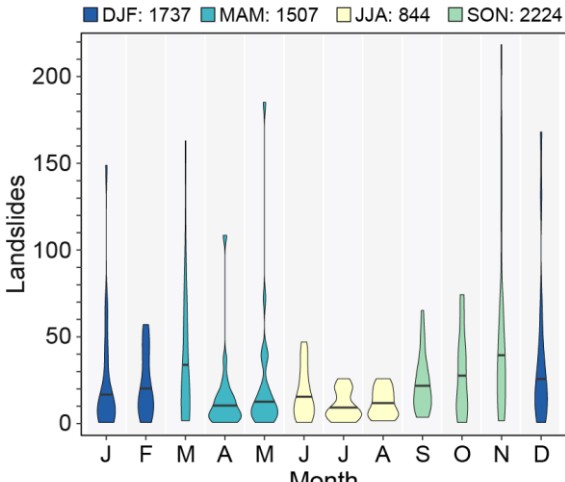

**Figure 3. Violin plot of the monthly distribution of landslides. Months are grouped into four seasons: DJF (December–January–
February), MAM (March–April–May), JJA (June–July–August), SON (September–October–November). The number of landslides
in each season is shown in the legend.**





**Table 4. Monthly statistics of number of landslides**

| month | Number of landslides | | |
|---|---|---|---|
| | **median** | **mean** | **total** |
| **Jan** | 11 | 23.67 | 544 |
| **Feb** | 15.5 | 22.9 | 457 |
| **Mar** | 16 | 35.3 | 670 |
| **Apr** | 7.5 | 15.6 | 312 |
| **May** | 10 | 23.9 | 525 |
| **Jun** | 12 | 17.3 | 363 |
| **Jul** | 7.5 | 10.6 | 234 |
| **Aug** | 11.5 | 12.4 | 247 |
| **Sep** | 17 | 23.4 | 491 |
| **Oct** | 19 | 28.6 | 658 |
| **Nov** | 30.5 | 48.9 | 1075 |
| **Dec** | 20 | 35.0 | 736 |

Figure 4 plots the number of landslides in each month in the entire observation period January 1996–December 2021. The
mean and median values of the annual number of landslides are reported. The first six years show much lower monthly and
annual values than the following 20 years: only 65 landslides were collected between 1996 and 2001. Overall, large variations
are observed between monthly and annual values. In particular, four years, namely 2008, 2009, 2010, and 2014, have
significantly higher mean and median values than the other years. Four years, from 2007 to 2021, are characterised by less
intra-annual variability, thus displaying very similar mean and median values.



**Figure 4.** Monthly distribution of landslides in the observation period January 1996–December 2021. The mean and median number of landslides per each year are also shown.

Figure 5 shows the number of landslides with different levels of geographic and temporal accuracy (see Table 1 for descriptions) over the entire observation period 1996–2021, and two sub-periods 2002–2011 and 2012–2021. The number of landslides collected each year is also depicted. About half of the landslide records (3131) have a high geographic accuracy (Fig. 5a) and more than 95% (6026) of the landslides were located with an uncertainty of less than 10 km$^2$. Only 285 (4.5%) landslides have low and very low geographic accuracy. On the other hand, for almost half of the catalogued landslides (3068)



the exact time of occurrence is known (Fig. 5b). For another quarter of the catalogue, the part of the day is known. Only 22%

of the landslides (1405) are characterised by a lower temporal accuracy.

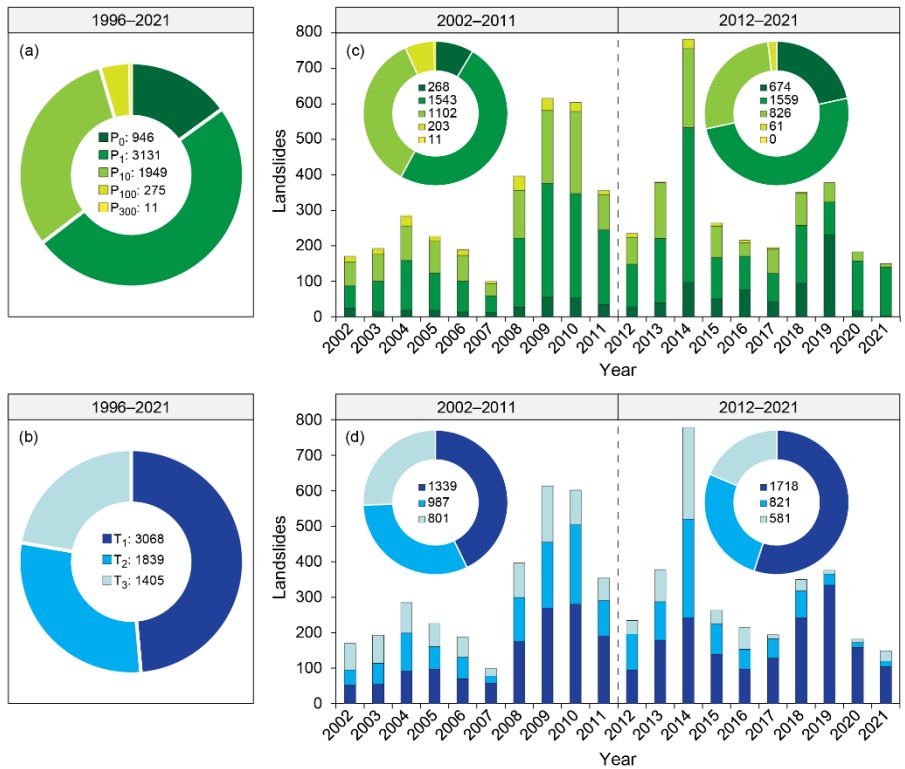

**Figure 5. (a,b) Donut charts of the number of landslides with different levels of geographic (P$_0$, P$_1$, P$_{10}$, P$_{100}$, P$_{300}$) and temporal accuracy (T$_1$, T$_2$, T$_3$) in the observation period 1996–2021. (c,d) Bar charts of the annual number of landslides for the different**

**geographic and temporal accuracy levels over the period 2002–2021. The donut charts in the insets show the comparison of the number of landslides in each class in two sub-periods 2002–2011 and 2012–2021. Refer to Tables 1 for a description of the geographic and temporal accuracy levels.**

We excluded the period 1996-2001 from the analysis due to the small (not representative) number of events (Fig. 4) and

identified two 10-year sub-periods, 2002–2011 and 2012–2021, which present a comparable number of landslides, 3127 and 3120 respectively. Figures 5c,d show that both geographic and temporal accuracy have improved in the most recent period. The number of landslides with very high geographic accuracy more than doubled from 2002–2011 to 2012–2021 period (Fig. 5c). Specifically, in the recent 2012–2021 period, the location of about three quarters of the listed failures is known with an accuracy of less than 1 km$^2$ and the time of occurrence with an uncertainty of less than 2–3 hours. In addition, the number of

landslides for which only the day of occurrence is known (T$_3$) decreased from 25% to 19% of the total (Fig. 5d).





Overall, more than a third of the catalogue (2175) is highly accurate both in space ($P_0$ and $P_1$) and time ($T_1$), whereas less than 2% of the catalogue (114) has concurrently low and very low ($P_{100}$ and $P_{300}$) geographic accuracy and daily ($T_3$) temporal resolution. These last records were collected mainly in the first years of the catalogue's compilation.

Figure 6 shows the subdivision of the records according to the source of information: institutional reports (IR) or news (NR).

Overall, 58% of the landslides were catalogued thanks to the information gathered from news reports. The same figure also shows how the information source affects the geographic and temporal accuracy of the landslide records. Among all 946 landslides having very high geographic accuracy, 766 (81%) were catalogued from information in institutional reports. On the other hand, 75% (2286 out of 3068) of all landslides with very high temporal accuracy ($T_1$) came from institutional reports. Both geographic and temporal accuracy substantially decrease when the landslides information is collected from news reports.


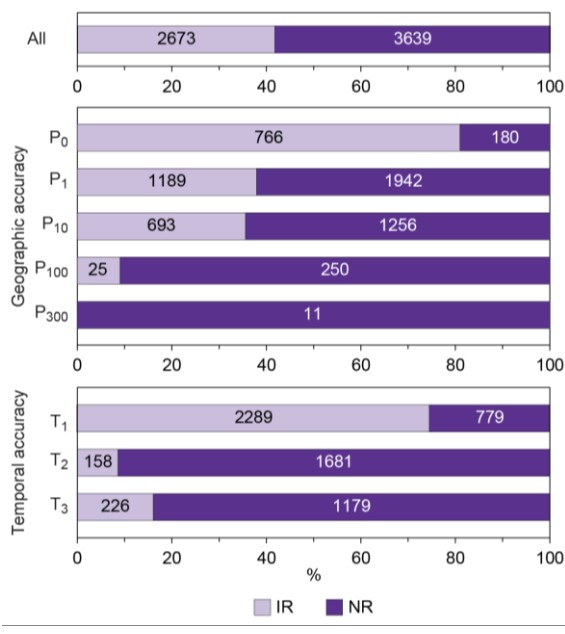

**Figure 6. Bar chart of the number of landslides whose information was gathered from institutional (IR) or news (NR) reports, divided in the classes of geographic and temporal accuracy.**

## 6 Data availability

ITALA catalogue is available at https://zenodo.org/record/7646106#.Y-4I_HbMJPY (Brunetti et al., 2023).



# 7 Remarks and conclusions

ITALICA is the largest catalogue of rainfall-induced landslides accurately located in space and time currently available in Italy. In particular, in the lowest geographic accuracy level $P_{300}$ the location of the landslide may be inaccurate for less than 10 km-radius. Similarly, the worst temporal accuracy still requires knowing at least the day of occurrence of the landslide which

is the habitual maximum accuracy in most of the catalogues. These two constraints, combined with the requirement for specific reference to rainfall as the sole triggering factor of the landslides, significantly limits the number of events suitable for the catalogue. As a result, on average, 40% of the analysed news reporting items are discarded. These strict and stringent criteria limit the use of many technical reports prepared in the aftermath of severe weather events in which generally only the date of the inspection and not the date of occurrence of the landslide is reported. In this regard, particularly useful are the reports

provided by firefighters, which are accurate both in space and time. Unfortunately, the availability of such data is not uniform across the territory due to the different data sharing policy of provincial and regional authorities. We noticed how both geographic and temporal accuracy increase substantially when information on landslides is gathered from institutional reports (Fig. 6). We can therefore state that a higher availability of such data sources would result in a more thorough catalogue.

Gathering information on rainfall-induced landslides that is accurate in both space and time requires a large amount of human

resources and time. For example, in order to get an accurate spatial location of landslides along roads, we explored the area using Google Street View until we found the proper sites, recognizable either through the images accompanying the information source or from the mileage information. The collection of such information can only be partially automated as expert supervision remains necessary.

ITALICA's elevated spatial and temporal accuracy is its main strength. The demand for accuracy in landslide catalogues is

critical for effective and operational landslide prediction through rainfall thresholds (Segoni et al., 2018; Guzzetti et al., 2020, 2022). Without this high level of accuracy, the rainfall responsible for triggering landslides cannot be reliably reconstructed. Another strength of the catalogue is that the collection was made according to strict objective and homogeneous criteria throughout the Country, which limits the inherent subjectivity in compiling catalogues from different sources and by multiple operators.

As with other landslide catalogues available in the literature (e.g., Kirschbaum et al., 2010, and references therein), this catalogue also exhibits some spatial inhomogeneity (Fig. 2), which may have implications for users. The main reason is attributable to the regional scale at which the search for information was conducted as a result of collaboration agreements with some Italian administrative regions (i.e., Liguria, Marche, Sardinia). For instance, the catalogue lists information on 1703 (27%) and 1220 (19%) rainfall-induced landslides in the Liguria and Marche region, respectively. A possible functional

definition of completeness requires that a historical landslide catalogue includes a substantial number of landslides in any year. According to this definition, ITALICA is substantially complete for the regions Liguria, Marche and Sardinia, for which there has been continuity in data collection from 2012 onwards. Despite the inhomogeneity of the data, ITALICA provides sufficient numbers of rainfall-induced landslides at the scale of administrative regions (Fig. 3b), which can be used for landslide





prediction models. As an example, as stated in Peruccacci et al., (2017), reliable rainfall thresholds for the possible landslide initiation can be defined in areas where the number of available records is larger than or equal to 100. The catalogue is continuously being updated, and future collaboration with other Italian regions will hopefully increase the spatial homogeneity of the data.

Compared to other catalogues of landslides in Italy, ITALICA stands out because: (i) it contains exclusively landslides induced by rainfall, contrarily to Guzzetti et al., (1994), Innocenzi et al., (2017), Calvello and Pecoraro, (2018), which contain information on landslides triggered by all causes; (ii) it covers a longer period (26 years, 1996–2021) than that covered by Innocenzi et al., (2017) (4 years, 2012–2015) and Calvello and Pecoraro, (2018) (8 years, 2010–2017); (iii) it is based on both technical and chronicle sources of information; and (iv) it is highly accurate both in space and time.

ITALICA cannot be compared with the IFFI inventory, as the two products are structurally different. The IFFI inventory includes landslides induced by multiple causes (e.g., anthropogenic, seismic) and mostly lacks information on the temporal occurrence of the mapped failures. On the other hand, a comparison with Polaris (Bianchi and Salvati, 2023) and ISPRA (Trigila et al., 2021) catalogues is not advisable, since the two contain information on landslides induced by all types of triggers that caused deaths, injuries, evacuations, and damage. As a matter of fact, in the four-year period 2017–2020, ITALICA lists 1102 landslides, while the Polaris and ISPRA catalogues contain 94 landslides with deaths and injuries and 645 landslides with deaths, injuries, evacuations, and damage, respectively.

Global and continental catalogues available in literature report a scarce number of landslides occurred on the Italian territory, i.e. 45 landslides in the period 2008–2013 according to Kirschbaum et al., (2015); 72 landslides in 2005–2014 according to Haque et al., (2016); 39 in 2004–2016 according to Froude and Petley, (2018). As expected, a national catalogue is certainly more comprehensive than the global catalogues in the area of interest addressed.

Recently, projects have been launched that involve citizens in providing reports of natural disasters that take lives and destroy roads, buildings, and other property, such as Landslide Reporter, a NASA citizen science project that asks citizen scientists from around the world to report landslides near them, allowing continuous feedbacks from the real world (https://gpm.nasa.gov/landslides/index.html). In the near future, we plan to use similar initiatives. Additionally, the usefulness of social media data is being tested and seems promising, suggesting their possible future integration in a multiple information source catalogue (Franceschini et al., 2022a, b).

In general, ITALICA's information on rainfall-induced landslides in Italy, with special emphasis on their spatial and temporal location, can be crucial for decision-making in landslide risk management. The methodology used to populate ITALICA has already been applied in a standardised way by various operators in different geographic and climatic contexts of Italy and can be easily used to compile new catalogues highly accurate in space and time in other countries.

**Author contribution**

SP, MTB: conceptualization, data curation, formal analysis, investigation, supervision, visualization, writing - original draft preparation, review & editing. SLG, MM: conceptualization, data curation, formal analysis, investigation, visualization, writing - original draft preparation, review & editing. MS: data curation, investigation. FG: funding acquisition, writing - review & editing.

**Competing interests**

The authors declare that they have no conflict of interest.

**Acknowledgements**

Work financially supported by the Italian National Department for Civil Protection (DPC) (Intese Operative DPC n. 619, 672, 1015, 1181; Accordi di Collaborazione 2014, 2015, 2016), environment department of the Liguria region (Convenzione 2013), the Apulia Region (Accordo di Collaborazione 2016) regional agency for the protection of the environment of the Liguria

region (Accordi di Collaborazione 2017, 2018, 2021), civil protection department of the Sardinia region (Accordi di Collaborazione 2016, 2021). Devis Bartolini, Francesca Brutti, Cinzia Bianchi, Costanza Calzolari, Barbara Denti, Eleonora Gioia, Silvia Luciani, Maria Elena Martinotti, Michela Rosa Palladino, Luca Pisano, Anna Roccati, Monica Solimano, Carmela Vennari, Giovanna Vessia, and Alessia Viero contributed to collect landslide information. We thank the functional centre for civil protection of the Marche region, provincial commands of the national fire and rescue service, *Centro Coordinamento*

*Informazioni Sicurezza Stradale*, *Rete Ferroviaria Italiana* for providing landslide information.

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
