# Peer review of "ITALICA, an extensive and accurate spatio-temporal catalogue of rainfall-induced landslides in Italy"

_Earth System Science Data, 2023_

## Referee Comment (RC1)

**Review of manuscript ESSD-2023-61:**

**"ITALICA, an extensive and accurate spatio-temporal catalogue of rainfall-induced landslides in Italy"**

**By:   Silvia Peruccacci, Stefano Luigi Gariano, Massimo Melillo, Monica Solimano, Fausto Guzzetti, Maria Teresa Brunetti**
* * *
**1) General comments**

Dear Editor, dear Authors

This manuscript by Peruccacci et al. presents a very valuable, high quality dataset on rainfall induced landslides in Italy. These data are crucial for the development of warning systems based on empirical rainfall thresholds with high spatial resolution. In addition, inventories of landslide events are an important cornerstone for improving scientific understanding of the processes involved and for supporting hazard assessment. This article shows well how time-consuming it is to compile such a database, a fact that tends to be forgotten by the general public (and sometimes, unfortunately, by users). The contribution represents a milestone for the development of future landslide early warning systems in Italy and it is to be expected that it will be widely cited in the future (together with the actual dataset by Brunetti et al., 2023).

The article is (with a few exceptions) well-structured and quite well written, the length of the text is adequate and the explanations and statements are well illustrated. Below I list three comments of a more general nature and ask the authors to comment briefly. In the second part of my review, I go through the individual sections/chapters of the article. The specific comments that arise are not of a fundamental nature and should be relatively easy for the authors to answer. They reflect questions I asked myself while reading or text passages I stumbled over. Finally, the third part contains a fairly extensive list of technical details that I encourage the authors to consider carefully.

- As can be seen in Figure 4 and made clear in lines 297-298, only a few events could be recorded for the first six years of the study period (65). Of course, this can have various reasons. It may be that only a few landslides really occurred in Italy in these years (which is probably rather unlikely), or the data basis for these early years was significantly worse than for the following years. Especially if the latter is the case, the question arises as to why 1996 was chosen as the start of the study. In my opinion, this is not explained in the text and would be interesting. Please also note my comment on Figure 4 in section (2) of this review.

- It is implicitly clear from the text in lines 185-188 as well as from Table 1 that triggering rainfall (as well as its reconstruction) is not part of ITALICA (or is not part of the data collection to feed ITALICA). This is probably due to the fact that a central elaboration of the triggering rainfall amounts for all events is very time-consuming and this step is probably done at a later stage by the landslide risk managers (for a selection of events that are of interest e.g. for a province or a region). For this article this information is not crucial, nevertheless I would briefly state/describe this fact in section 4 (just before or after table 1) in one or two sentences.

- Both the Abstract and the Concluding Remarks point out the importance of spatially and temporally accurate information on landslide events for their use in early warnings in Italy. Are there already applications that demonstrate or support these statements and could be briefly mentioned in the Concluding Remarks? If not, do the authors know if such applications are planned that could be mentioned? If there is something more detailed to report here, a short additional section on "Research Opportunities" or "Research Applications" would be quite conceivable and would further strengthen the contribution.

In summary, this is an important contribution that will be of considerable interest to the landslide research community as well as to local and regional decision makers in landslide risk management. In my opinion, it should therefore be published in ESSD. I propose that the paper be accepted subject to minor revisions.

**2) Specific comments on the different sections of the article**

**Introduction**

The introduction is well and concisely written and prepares the reader nicely for the following chapters. The only thing I miss is a short definition of the aims of this paper at the end of the last paragraph. Something in the spirit of: "The aim of this publication is to present this dataset to the scientific community as well as natural hazard managers so that they can apply our data or build similar databases elsewhere […]".

L35:  I suggest adapting this sentence to make it clearer:
"Between 1972 and 2021, landslides made 145,548 people homeless or evacuated and caused 2504 casualties…"

L40-42:  I strongly recommend not making paragraphs that consist of only one sentence. I think the paragraph mark in line 42 can be removed.

L47-48:  Is there a reference to support the statement that rainfall thresholds are good for forecasting? For example, one of the references listed in line 52.

L53:  What exactly is meant by "local regional thresholds"? I would have thought the order (with increasing spatial resolution) would be regional threshold → sub-regional threshold → local threshold.

**Background**

Overall, section (2) Background is currently the weakest in my opinion. Not in terms of content (and I would like to emphasize that such an overview is generally very helpful), but in terms of organization. The section is a little long and has a somewhat confusing structure. In the current version, no system is clearly evident in these five paragraphs. The order of the studies/databases described does not seem clear to me, and there are jumps from one country to another. I wonder if compiling an overview table with the key information on the different databases, inventories and catalogues could be helpful for readers.

At the very least all information on catalogues, databases and inventories from Italy should be bundled in the text. Such a section on Italian efforts could be concluded with the important statement in lines 134-135, which stresses the advantages of ITALICA, described in detail in the following chapters (4) and (5).

The different national databases described cover mainly Europe. Outside Europe, data collections for Nicaragua (lines 72-75) and New Zealand, (lines 99-103) are mentioned. Presumably there are other inventories outside Europe that would be worth noting. For example, the U.S. Landslide Inventory (https://www.usgs.gov/tools/us-landslide-inventory) of the USGS comes to mind, which should certainly also be referred to. However, this means that section 2 of the MS would become even longer. This could be an argument for using a list.

L63-65:  I am not sure whether the definition of the terms "catalogue", database" and "inventory" (in lines 63-65) is really necessary. Do the authors want to point out that these three terms are often used synonymously in the literature? If so, I would formulate this more directly. In your opinion, would synonymous use be correct or incorrect?

L86:  The information on the 17000+ records of landslide events recorded by the British Geological Survey would be much more valuable if the period of investigation (from year to year) were specified. Add if possible.

L88:  See immediately above: here, too, it would be interesting to know the number of years covered for landslide data collection in Poland (~40000 entries in how many years?).

L96: See immediately above: here, too, it would be interesting to know the number of years covered for landslide data collection in Slovenia (6234 entries in how many years?).
The same comment applies to the information on New Zealand in line 101.

L104-05: Here I don't understand something in relation to the database of Innocenzi et al. The authors write that of 1054 events only 808 had geographical information (and 246 therefore had none). However, the same sentence begins with the statement "Each landslide was assigned a location". Something can't be right. Please clarify.

L125: Please check the work of Kirschbaum et al. (2010) again. If I remember correctly, they did not compile worldwide landslides of the years 2003 to 2008, but of the three years 2003, 2007 and 2008.

**Study area**

The section is concise and precise. I have only one small question/issue. According to Figure 2a, the vast majority of landslides occur in the Alps and the Apennines (these two mountain ranges are described in section 3. Furthermore, the islands/regions of Sicily and Sardinia, as well as the region of Calabria, show numerous events. (a) Would it be worth mentioning in line 143 that Sicily and Sardinia are hilly/mountainous? (b) Do the hills and mountains of Calabria count as part of the "Appennino meridionale" or should something additional be mentioned here very briefly in the text?

Figure 1: I suggest extending the caption a little. For example, how about:
"Study area: the Italian peninsula. Background from Bing; EPSG: 4326." Or:
"Study area: The peninsula of Italy, including the two main islands Sicily and Sardinia. Background from Bing; EPSG: 4326."

**Data and methods**

This is not absolutely necessary, but as section 4 is somewhat longer than all the others, subtitles might be helpful here. E.g. 4.1 Structure, 4.2 Information source, 4.3 Landslide type, 4.3 Spatial information, 4.4 Temporal information, 4.5 Organization and quality control.

L173-74: By "uneven" do the authors mean "inconsistent"? And right after that, by "conflicting" do they mean "contradictory"?

L185: Consider rephrasing to "The catalogue contains the following information for each record"

L187: "Date" is the combination of "day, month, year". It is OK to differentiate in Table 1; here in the parenthesis, however, the specification of "date" is rather confusing.

Table 1: Consider expanding the caption slightly as follows:
"Summary of fields included in the ITALICA catalogue."

L192: Since immediately above this text passage (in lines 185-187) an enumeration (i) to (iv) is already made, I think the enumeration (i) to (ii) can be omitted here. It is not really necessary.

L193: Consider changing to "news websites,"

Table 2: It is obvious, but perhaps it should still be pointed out in the caption that all the terms in Table 2 have been translated from Italian.
Are the original key terms (in Italian) available somewhere (e.g. supplementary material)? That would be a nice-to-have, but is not absolutely necessary.

L214: While the adjective "reliable" fits well, the adjective "accurate" seems a bit strange to me here. "Accurate information about the exact or approximate landslide location", does that make sense? I would change it to: "...provide reliable information about the exact or approximate landslide location"

| L220-22: | I don't quite understand this sentence (or rather the last part of it). If the sources lack a precise description of the process, how can an event be assigned to one of these five types? Is the most likely process simply assumed for the given location? Please check sentence again. |
| --- | --- |
| L236: | I am not sure what the authors mean by "sighted"? Wouldn't "visible" be better? Is it meant here that a landslide was visible in Street View (if not, a new sentence would have to be started, instead of semicolon). |
| L237: | I suggest deleting the parenthesis "(i.e., 1 km$^2$ or less)". This specification is clearly stated in line 230 and in Table 1 and is not necessary again here. |
| L241: | From line 232 to line 241 it is clearly explained when $P_0$, $P_{10}$ or $P_{100}$ were attributed respectively. According to Table 1, however, there is also $P_{300}$. For the sake of completeness, perhaps $P_{300}$ should also be mentioned briefly? |
| L244: | The text regarding $T_2$ is a bit awkwardly worded; or more specifically, I stumbled across the verb "inferred". Would the following suggestion be correct in terms of content?
"T2 when the part of the day is known or the time can be estimated reasonably well." |

**Description of the catalogue**

| L266-67: | This sentence can be misunderstood. The higher number of events results because the local authorities assisted in obtaining data, right? If necessary, rephrase, e.g.:
"Some areas have a higher concentration of events due to special agreements with local authorities for data collection." |
| --- | --- |
| L273: | Change to "were recorded" or "information on more than 1000 landslides was collected" |
| L283-84: | Regarding the statement: "The difference between the mean and the median values is significant for all months,"
I disagree for the month of August, where the difference is very small. This should be noted here, for example in brackets. |
| Figure 2: | Consider the following two minor additions to the caption:
"Location of the catalogued landslides in ITALICA,"
"The number of landslides of each type is given in brackets in the legend." |
| Figure 3: | Consider slightly rephrasing the caption as follows:
"Violin plot of the monthly distribution of landslides in ITALICA. Months are grouped into four seasons: Winter DJF (December–January–290 February), spring MAM (March–April–May), summer JJA (June–July–August), and autumn SON (September–October–November). The number of landslides in each season is shown in the legend." |
| Table 4: | The data set covers 26 years. Strangely, however, the mean values of the individual months in table 4 were not calculated using the 26 years. For example: January has a total of 544 landslides; 544/26= 20.9; however, 23.67 is given as the "mean"; this would correspond to an investigation period of 23 years with 544 events.
None of the mean values of the months in Table 4 seem to have been calculated on the basis of 26 years (this varies between 19 years in March and 23 years in January and October. |
| L296-97: | I think this sentence does not quite describe Fig. 4 correctly. My suggestion:
"For each year, the mean and median values of the monthly number of landslides are given in the corresponding panel." |
| Figure 4: | The first six years of the study period show hardly any events. This observation is noted in lines 297-298 by the authors. In fact, the catalogue averages around 11 events per month for the first six years (1996-2001) and around 312 events per year for the following 20 years. That is a massive |

difference. Is this solely due to the weather (i.e. hardly any heavy precipitation events triggering landslides) or is there also a methodological reason (i.e. that the data collection in later years allowed the identification of more events). I think it would be good if the authors could briefly explain or discuss this difference somewhere in the text.

L300:     Instead of "from 2007 to 2021" I suggest "between 2007 and 2021"
          Also, it is not entirely clear which four years the authors mean; presumably 2007, 2017, 2020 and 2021. Median and mean values are also close to each other for 2015 and 2016 (with rather low intra-annual variability). Perhaps this could be made more precise.

L321-23:  I feel that this important information should be stated before the authors begin to describe the well-illustrated Figure 5. Consider moving this sentence before line 306.
          Please also refer to Figures 5c and 5d in this sentence, as the exclusion of the 1996-2001 data only affects this figure (as far as I understand). E.g. "We excluded the period 1996-2001 from the analysis in Figures 5c and 5d…"

L328:     It is not clear whether this statement ("Overall, more than a third of the catalogue (2175) is highly accurate...") refers to the entire catalogue or to the reduced catalogue (2002-2021). Please clarify.

Figure 6   In the current version of the figure, the information content was always assigned to only one source (IR or NR). Presumably this is the "main source". However, in lines 215-217, the authors mention that the temporal and spatial accuracy could often be improved by combining institutional reports with news reports. Would there be a way to show this in Figure 6?

**Remarks and conclusions**

Consider renaming this section "Concluding remarks" because for a classic/typical chapter "Conclusions" it is a bit lengthy (compared to the length of the whole text).

L344-45:  Consider changing to the more straight forward
          "…of rainfall-induced landslides with precise spatial and temporal localisation currently available…"

L345-47:  In lines 345-347, I can understand the authors' reasoning. However, I find the two sentences not optimally formulated. I make an alternative wording suggestion below and let the authors decide how they want to proceed.:
          "The selection criteria are relatively strict compared to other inventories. At the lowest acceptable geographic accuracy level, P300, the location of the landslide can still be indicated within a radius of less than 10 km. And the worst level of temporal accuracy, T3, still requires that at least the day of occurrence of the landslide is known, which is the usual maximum accuracy in most catalogues."

L361ff:   The paragraph from lines 361 to 366 is still about spatial and temporal accuracy. In my opinion, it should therefore be attached directly to the first paragraph (after line 355). The statements regarding required manpower and time (now lines 356 to 360) can follow afterwards.
          I would also make the first sentence more concise:
          "The high spatial and temporal accuracy of ITALICA is its main strength."

L370ff:   I do not understand the reasoning of the authors in lines 370-374. Are they saying that the database is only temporally homogeneous (complete?) for those administrative regions with which collaboration agreements exist and not for all other regions?
          The statement about "completeness" also made me wonder. Isn't it possible, for example, that in a very dry year in one region only very few landslides occur and thus only a few events can be recorded?

**3) Technical details**

L32:   Consider "disasters" instead of "catastrophes"

L33:   Consider changing to "a variety of causes, including…"

L33:   Consider changing to "…and anthropogenic factors."

L42:   Change to "rainfall-induced landslides"

L43:   Consider changing to "over large areas"

L45:   Consider changing to "…between rainfall and landslide occurrence,"

L47-48:  I suggest changing to:
"…predicting the occurrence of shallow landslides, where there is a direct correlation between rainfall and landslide initiation. However, rainfall thresholds are…"

L57:   Consider changing to "…homogenous catalogue containing accurate information…"

L62-63:  Consider changing to "… in the literature, which are briefly reviewed in this paper."

L69-70:  Consider changing to:
"…as part of the national project AVI – *Aree Vulnerate Italiane* (an acronym for Areas Affected by Landslides or Floods, Guzzetti et al., 1994). Subsequently, …"

L71:   "Guzzetti (2000) compiled…" instead of "(Guzzetti, 2000) compiled…"

L72:   "expanded by Salvati et al. (2010, 2018)." instead of "expanded by (Salvati et al., 2010, 2018)."

L79:   Consider changing to "Hervás (2012) published a detailed analysis of the (then) existing national landslide databases in Europe."

L90:   Rather than "triggering events", I would prefer "triggering factors" or "event triggering" in this context.

L130:   Here and in lines 266, 271 and 365 I would not capitalize the word "country"

L137:   Consider changing to "…covering 301336 km$^2$ in …"

L138:   Consider writing "mountain ranges" instead of "mountain chains"

L139:   Consider changing to "…west-to-east arc across the northern tip…"

L139-40:  Change to "…from east to west reaching an altitude of over 4,800 m a.s.l. and separating the Italian peninsula…"

L140:   Change to "The Apennines are a…"

L141:   Change to "Elsewhere, Italy plunges into the Mediterranean Sea and is surrounded in particular by the Adriatic,"

L147:   Consider changing to: "as it lies at the boundary between the Eurasian and African plates."

L150:   Change zo "In the north…"

L151:   Change to "…increasing towards the south."

L151-52:  Change to "…and prealpine areas as well as the northern Apennines have higher precipitation, with mean annual values exceeding 2,000 mm."

L155:   Consider removing the paragraph mark (avoid single-sentence paragraphs)

L162:   Consider changing to "…occurred in uninhabited areas or for which there are no institutional or news reports are rarely included in the ITALICA catalogue"

L164:   Change to "In order to obtain detailed and up-to-date…"

L168:   Change to "…in archives at different levels of…"

| | |
|---|---|
| L168: | Consider using "(municipal, provincial and national)" to avoid confusion with the archives of the US states |
| L170: | Change to "…involvement of institutions…" |
| L179: | Consider changing to "…such as the general location and the date of occurrence…" |
| L194: | Change to "…systematic searches of regional…" |
| L205: | Please add comma after "social media posts" |
| L210: | Use full stop instead of comma: "…and State Forestry Corps." |
| L210: | Instead of "were provided" I would use "are provided by ANAS". Or does that statement only apply to the past? |
| L214: | Consider changing to "…have proved to be particularly useful," |
| L224: | Change to "Landslides are then represented as points…" |
| L228: | Consider changing to "which make it possible to consult all the maps…" |
| L234: | Change to "with the exact kilometric indication" |
| L236: | Consider changing to "in a few cases," |
| L249: | Consider changing to "…to determine the inferred time of failure," |
| L250: | Consider changing to "Lastly, if the news only reports the day…" |
| L251: | Change to "… a daily temporal accuracy of $T_3$ and is conventionally assumed to have occurred at the end of the day (23:59)." |
| Table 3: | Why is the inferred time for the "Night" time slot not 04:00, which is practically halfway between 23:59 and 08:00? At least in summer, 05:00 is almost the time of dawn. |
| L260: | Consider removing the paragraph mark (avoid single-sentence paragraphs) |
| L264: | Change to "…that occurred on the Italian territory…" |
| L275: | According to the text in line 270, the abbreviation for "mud flow" is MF (and not MD). Please clarify |
| L283: | Change to "…by a factor of 2.5," |
| L287: | Consider changing to "…are mainly located in the…" |
| L307: | Consider extending to: "…and two 10-year sub-periods 2002-2011 and 2012-2021." |
| L310: | Change to "…have a low and very low…" |
| L316: | In the figure caption, change text to "…in the overall observation period 1996-2021." |
| L318: | In the figure caption, change to "Refer to Table 1 for…" |
| L323: | Here and in line 325, replace "period" with the more accurate "sub-period" (introduced in line 322) |
| L324: | Change to "…has more than doubled from the sub-period 2002-2011 to the sub-period 2012-2021." or shorter "…has more than doubled from 2002-2011 to 2012-2021." |
| L327: | Consider changing to "…of the total events (Fig. 5d)." |
| L329: | Consider extending to "…and only daily ($T_3$) temporal resolution." |
| L339-40: | In the figure caption, consider changing text to "…number of landslides (1996-2021) for which information was obtained from institutional (IR) or news (NR) reports, divided into classes of…" |
| L348: | Change to "…significantly limit the number…" |
| L349: | Consider changing to "As a result, an average of 40%..." |
| L350: | Use "produced" instead of "prepared" |
| L353: | Consider using "country" instead of "territory" |

| L353: | "policies" |
|---|---|
| L354: | Consider "is collected" instead of "is gathered" |
| L368-69: | Consider changing to "The main reason for this is the regional scale at which…" |
| L371: | The percentage refers to all entries in ITALICA; this would need to be specified, e.g. "(27% of all entries)". |
| L381: | Consider adapting to "it contains exclusively landslides induced by rainfall, unlike datasets presented in Guzzetti et al. (1994), Innocenzi et al. (2017), Calvello and Pecoraro (2018), which contain information on landslides…" |
| L387: | Change to "…a comparison with the Polaris…" |
| L392: | Consider changing to "…available in the literature report a small number of landslides on the Italian territory," |
| L393: | Change to "…according to Kirschbaum et al. (2015); 72 landslides in the period 2005–2014 according to Haque et al. (2016); and 39 in the period 2004–2016 according to Froude and Petley (2018)." |
| L398: | Change to "to report landslides in their area, providing continuous feedback from the real world" |
| L400: | Change to "…suggesting its possible future…" |
| L404-05: | Consider changing to "…and climatic contexts in Italy and can easily be used to compile new catalogues of high spatial and temporal accuracy in other countries." |

---

## Referee Comment (RC2)

**Review of manuscript ESSD-2023-61: "ITALICA, an extensive and accurate spatio-temporal catalogue of rainfall-induced landslides in Italy" By: Silvia Peruccacci, Stefano Luigi Gariano, Massimo Melillo, Monica Solimano, Fausto Guzzetti, Maria Teresa Brunetti**

Dear Editor and Authors.

Thank you for your valuable and detailed presentation and examination of the ITALICA dataset. I feel this is a good paper that clearly presents the difficulties, resources, and data variability available in compiling such an inventory. The focus of the work is clearly defined, and the steps/ methodologies adhered to in order to compile such a dataset are clearly established.

I found the paper to be engaging and comprehensive, with questions arising mostly being answered as the paper progressed. The Figures and Tables all support the text and are positioned well to support the narrative.

I have a few very minor comments that I would like to see addressed to support the paper, most of which appear to fit with the Background section. Please see comments supplied.

I would recommend this paper be accepted for publication with very minor edits.
* * *
**Section 2: Background**

**Lines 66 – 133** Whilst presenting a good background of inventories, could this section be restructured slightly either as a list or by geographic area. E.g. Italian databases are mentioned at the beginning and the end, Nicaragua in between two Italian databases. May I suggest reviewing this section and formatting along the lines of *Italy > Europe> rest of the world example > Global*.

- o I would suggest that the authors review if they are stating examples or offering comparisons of contents.
- The text should clearly define the time limited databases and those which are still being updated. This is not always well presented.
  - o L84 Change "included" to "to date includes" – the GB database is being updated constantly and is not a static inventory. It also is different to some of the other databases presented as it includes historic (preglacial) undated landslide event deposits not just present day (hence 17000)
  - o L85 Poland details until April 2014 but from when – is this time stamped.
  - o L94 Slovenia – ongoing but from when.
- Line 84 suggest an edit "with capability to include" – data is only added where it is available although the fields are present as stated in references.

**Section 4: Data and Models**

**Table 1** Geographic accuracy, the text is described "as geographic accuracy based on the area which the landslides realistically occurred described as radius from the coordinates" however the examples are presented as area. The approximate radius is mentioned later in the text (L235-239) but it would be helpful to the reader if the radius could also be included in the table along with the area to assist interpretation.

**Section 7: Remarks and conclusions**

**L375**    I would suggest the authors consider L375-377 "the catalogue…" should be reordered to the end of the paper as the introduction line to the beginning of the summary for the future of the inventory and follow on work.   and included in the final paragraph beginning L394.

**Technical notes:**

There are many areas where there are single sentence paragraphs. I strongly suggest these are reviewed and removed.

L30    "catastrophes" change to disasters

L148     "along the length of the peninsula"

L160    "non- anthropized areas" suggest "areas not subjected to any anthropogenic change".

L394    Natural disasters – This can be a controversial term in some current natural hazard/ social science circles. If authors agree I would strongly suggest change to "natural hazards". This is supported in the contact of the sentence.

L308    Colloquial "on the other hand" should be removed.

L332    Colloquial "on the other hand" should be removed.

L273    Caption for Figure 2 mud flow abbreviation is presented as MD – presented as MF in table 1 and L219 and L268. Please check.

L385    Colloquial "on the other hand" should be removed.

---

## Referee Comment (RC3)

Dear Editor, dear Authors

The manuscript by Peruccacci et al. highlights how essential reliable landslide datasets are in the elaboration of landslide thresholds, to be used in the development of landslide early warning systems. The authors also emphasize how landslide inventories are important tools in the hazard characterization and assessment and, they provide a very useful and quite up-to-date overview of existing landslide inventories around the world.

Despite some few examples, most of the landslide inventories have been prepared after the year 2000 and were created with the purpose to show where historical landslide events have occurred and the damages they have caused. In the compilation of these inventories, the main concern was mainly the quantity of data rather than the quality, or information about triggering mechanisms. Most of these inventories were prepared as part of specific projects and created during the duration of the project, not always updated, with new entries or improved in their quality, in later years.

The authors highlight quite well the difficulties in the compilation of landslide inventories and the fact that this is a time-consuming task. And this is, in my opinion, the reason why the compilation of landslide inventories is often a forgotten task in the national framework for landslide risk assessment, with very limited resources and personnel assigned, well indicated by Reviewer 1.

We must also remember that in the past many previous authors have prepared landslide thresholds by using separated landslide dataset. It is with the most recent need of operational landslide early warning systems that we demand reliable landslide data to use in thresholds analyses.

The authors described well the existence of other databases in Italy, their limitations and the need for a "separate" dataset for rainfall-induced landslides. This may seem a good solution at the beginning, but in a long run it will create problems for those who manage landslide prevention at national level. Therefore I have some general comments for the authors that I would like to be adressed in the document.

- It is not clear if and how ITALICA communicate with the other existing inventories. If a new rainfall-induced landslide should be registered in ITALICA, will this be send automatically in the other databases and viceversa? How to deal with future updates? Who is responsible for these updates?
- Have you considered the possibility to improve one of the existing databases instead of making a new one? Adding for example a "quality level code" or create a separate module with specific parameters for those landslides  that should be used for thresholds?
- Some operational early warning systems send alert messages also for landslide triggered by snow melt or only for high soil water saturation, what do you suggest based on your experience? do we need to create a separate database for each triggering?
- Many countries have one database, managed by one national institution, with limited resources and personal, and efforts are being made to improve the quality of the registered events, by assigning a quality level to each entry and to provide information

about triggering. Based on your experience and take into account the problems with future updates, what do you recommend? To create separate databases or improve the existing inventories?

- At the end you have created a database with both low and high quality data. Could you explain better if you could use the low quality data in your threshold analyses (maybe for threshold at regional scale requiring low accuracy)? or do you need always high quality data?

I also agree that the article is quite well structured and written, with adequate length and, with explanations and statements well illustrated.

It can be an important contribution and of interest to the landslide research community working in particular with landslide early warning systems. I would recommend this paper to be accepted for publication, with very minor edits (see supplied comments).

I have just few comments/edits on the text

- Page 2 line 46, "shallow landslides" maybe useful to add what kind of types are in this category
- Background: In the list of inventories, I have not seen this reference. *Herrera, G., Mateos, R.M., García-Davalillo, J.C. et al. Landslide databases in the Geological Surveys of Europe. Landslides* **15**, *359–379 (2018). [https://doi.org/10.1007/s10346-017-0902-z](https://doi.org/10.1007/s10346-017-0902-z)* . Please check if they have mentioned other inventories that are not in your review.
- Another reference that is missing is the one that describe the inventory for Norway: Jaedicke, C., Lied, K., and Kronholm, K.: Integrated database for rapid mass movements in Norway, Nat. Hazards Earth Syst. Sci., 9, 469–479, https://doi.org/10.5194/nhess-9-469-2009, 2009.

---

## Author Comment (AC1)

**Review of manuscript ESSD-2023-61:**

**"ITALICA, an extensive and accurate spatio-temporal catalogue of rainfall-induced landslides in Italy" By: Silvia Peruccacci, Stefano Luigi Gariano, Massimo Melillo, Monica Solimano, Fausto Guzzetti, Maria Teresa Brunetti**
* * *
**1) General comments**

Dear Editor, dear Authors

This manuscript by Peruccacci et al. presents a very valuable, high quality dataset on rainfall induced landslides in Italy. These data are crucial for the development of warning systems based on empirical rainfall thresholds with high spatial resolution. In addition, inventories of landslide events are an important cornerstone for improving scientific understanding of the processes involved and for supporting hazard assessment. This article shows well how time-consuming it is to compile such a database, a fact that tends to be forgotten by the general public (and sometimes, unfortunately, by users). The contribution represents a milestone for the development of future landslide early warning systems in Italy and it is to be expected that it will be widely cited in the future (together with the actual dataset by Brunetti et al., 2023).

The article is (with a few exceptions) well-structured and quite well written, the length of the text is adequate and the explanations and statements are well illustrated. Below I list three comments of a more general nature and ask the authors to comment briefly. In the second part of my review, I go through the individual sections/chapters of the article. The specific comments that arise are not of a fundamental nature and should be relatively easy for the authors to answer. They reflect questions I asked myself while reading or text passages I stumbled over.

Finally, the third part contains a fairly extensive list of technical details that I encourage the authors to consider carefully.

- As can be seen in Figure 4 and made clear in lines 297-298, only a few events could be recorded for the first six years of the study period (65). Of course, this can have various reasons. It may be that only a few landslides really occurred in Italy in these years (which is probably rather unlikely), or the data basis for these early years was significantly worse than for the following years. Especially if the latter is the case, the question arises as to why 1996 was chosen as the start of the study. In my opinion, this is not explained in the text and would be interesting. Please also note my comment on Figure 4 in section (2) of this review.
  You are right. We now added (line 296): "The limited number of landslides does not depend on particular weather conditions during the period, but on the fact that historical data were available for only a few regions."

- It is implicitly clear from the text in lines 185-188 as well as from Table 1 that triggering rainfall (as well as its reconstruction) is not part of ITALICA (or is not part of the data collection to feed ITALICA). This is probably due to the fact that a central elaboration of the triggering rainfall amounts for all events is very time-consuming and this step is probably done at a later stage by the landslide risk managers (for a selection of events that are of interest e.g. for a province or a region). For this article this information is not crucial, nevertheless I would briefly state/describe this fact in section 4 (just before or after table 1) in one or two sentences.
  We are already working on this topic and a publication will come out soon. We added the following sentence before Table 1: "The rainfall that likely triggered the landslides contained in the catalogue will be analysed in a forthcoming publication."

- Both the Abstract and the Concluding Remarks point out the importance of spatially and temporally accurate information on landslide events for their use in early warnings in Italy. Are there already applications that demonstrate or support these statements and could be briefly mentioned in the Concluding Remarks? If not, do the authors know if such applications are planned that could be mentioned? If there is something more detailed to report here, a short additional section on "Research Opportunities" or "Research Applications" would be quite conceivable and would further strengthen the contribution.
  Thank you for the suggestion. We prefer not to include a section on applications of the catalogue, but simply add the following sentence in the concluding remarks: "Subsets of the catalogue have been already used to calculate national and regional rainfall thresholds implemented in early warning systems in Italy (Guzzetti et al., 2020). ITALICA can also be exploited to calibrate and/or validate various models for temporal prediction of rainfall induced landslides."

In summary, this is an important contribution that will be of considerable interest to the landslide research community as well as to local and regional decision makers in landslide risk management. In my opinion, it should

therefore be published in ESSD. I propose that the paper be accepted subject to minor revisions.

R: We thank the reviewer for his valuable comments and suggestions.

**Specific comments on the different sections of the article**

**Introduction**

The introduction is well and concisely written and prepares the reader nicely for the following chapters. The only thing I miss is a short definition of the aims of this paper at the end of the last paragraph. Something in the spirit of: "The aim of this publication is to present this dataset to the scientific community as well as natural hazard managers so that they can apply our data or build similar databases elsewhere […]".

R: Thank you for the suggestion. We added the following sentence: "ITALICA provides the scientific community a useful example on how to build accurate spatio-temporal catalogues elsewhere."

L35:        I suggest adapting this sentence to make it clearer:

"Between 1972 and 2021, landslides made 145,548 people homeless or evacuated and caused 2504 casualties…"

R: Done.

L40-42:    I strongly recommend not making paragraphs that consist of only one sentence. I think the paragraph mark in line 42 can be removed.

R: Done.

L47-48:    Is there a reference to support the statement that rainfall thresholds are good for forecasting? For example, one of the references listed in line 52.

R: We added references to two review articles highlighting the role of thresholds in landslide prediction.

L53:        What exactly is meant by "local regional thresholds"? I would have thought the order (with increasing spatial resolution) would be regional threshold ☐ sub-regional threshold ☐ local threshold.

R: We thank the review for pointing this out: "local regional" was a typo. We deleted the word regional.

**Background**

Overall, section (2) Background is currently the weakest in my opinion. Not in terms of content (and I would like to emphasize that such an overview is generally very helpful), but in terms of organization. The section is a little long and has a somewhat confusing structure. In the current version, no system is clearly evident in these five paragraphs. The order of the studies/databases described does not seem clear to me, and there are jumps from one country to another. I wonder if compiling an overview table with the key information on the different databases, inventories and catalogues could be helpful for readers.

At the very least all information on catalogues, databases and inventories from Italy should be bundled in the text. Such a section on Italian efforts could be concluded with the important statement in lines 134-135, which stresses the advantages of ITALICA, described in detail in the following chapters (4) and (5).

The different national databases described cover mainly Europe. Outside Europe, data collections for Nicaragua (lines 72-75) and New Zealand, (lines 99-103) are mentioned. Presumably there are other inventories outside Europe that would be worth noting. For example, the U.S. Landslide Inventory (https://www.usgs.gov/tools/us-landslide-inventory) of the USGS comes to mind, which should certainly also be referred to. However, this means that section 2 of the MS would become even longer. This could be an argument for using a list.

R: Thanks for the comment. Originally, the background section was organized by listing chronologically the examples of landslide catalogues, inventories, and databases found worldwide. We acknowledge that this organization could be a bit confusing, therefore we agree on the effectiveness of presenting the review following a geographical order. We decided now to start form the global scale, followed by non-European cases, European examples, and finally Italian references. We would prefer to keep the text instead of replacing it with a table. The new text of the section is now completely re-arranged.

As suggested, we also added a brief description of the U.S. landslide inventory, as follows: "In USA, an openly accessible inventory of landslides has been prepared by the U.S. geological survey (Belair et al. 2022;

https://www.usgs.gov/tools/us-landslide-inventory). To date the inventory includes more than 121,000 landslide points and almost 55,000 polygons across the entire USA territory, with details on date (with varying accuracies, from the day of occurrence to an undefined period), fatalities (if any), degree of confidence, and source of information."

Reference: Belair, G.M., Jones, E.S., Slaughter, S.L., and Mirus, B.B., 2022, Landslide Inventories across the United States version 2: U.S. Geological Survey data release, https://doi.org/10.5066/P9FZUX6N.

L63-65:     I am not sure whether the definition of the terms "catalogue", database" and "inventory" (in lines 63-65) is really necessary. Do the authors want to point out that these three terms are often used synonymously in the literature? If so, I would formulate this more directly. In your opinion, would synonymous use be correct or incorrect?
R: Thanks for this comment, which allows us to clarify this point. By listing the definitions of the three terms we wanted to point out that they have different meanings and related uses. In our opinion, using them as synonyms in the literature is incorrect.

We clarify now this point by adding the following text: "The three terms are often used as synonymous, even if they have different meanings." […] "Generally, a landslide catalogue should contain temporal information on landslide occurrences and not necessarily include geometrical data. In contrast, a landslide inventory typically includes spatial and geometrical data, leaving out precise dates of occurrence. Therefore, catalogues can be used for temporal prediction of landslides, e.g., to calculate rainfall thresholds, whereas inventories are employed for spatial prediction of landslides, e.g. for susceptibility analyses. A landslide database can include both temporal and spatial information, although examples in the literature are somehow inconsistent in this regard."

L86:     The information on the 17000+ records of landslide events recorded by the British Geological Survey would be much more valuable if the period of investigation (from year to year) were specified. Add if possible.
R: This information is not available. We checked again the mentioned papers and we had personal communications with one of the authors.
We specify this point in the text, which now reads: "In Great Britain, a landslide database was developed by the British Geological Survey (Foster et al., 2012), relying upon a variety of sources including maps, other databases, reports, research theses, and newspaper articles. To date it includes over 17,000 records of landslide events with more than 35 attributes, with capability to include location, landslide size and type, trigger mechanism, damage, and material. The database is not temporally limited between set dates; it includes also historic (preglacial) undated landslide event deposits. Temporal information (occurrence date) is stored only when available, especially for recent events, for which social media are also used to collect temporal information (Pennington et al., 2015)"

L88:     See immediately above: here, too, it would be interesting to know the number of years covered for landslide data collection in Poland (~40000 entries in how many years?).
R: Also in this case the information is not available.

L96:     See immediately above: here, too, it would be interesting to know the number of years covered for landslide data collection in Slovenia (6234 entries in how many years?).
R: This information is not available in the Slovenia database. We had personal communications with colleagues from the Slovenian Geological Survey to check this issue. They stored temporal information only for a few landslides.

The same comment applies to the information on New Zealand in line 101.
R: Again, in this case the information on the timing is not always available, as stated by the authors of the mentioned paper.

L104-05:     Here I don't understand something in relation to the database of Innocenzi et al. The authors write that of 1054 events only 808 had geographical information (and 246 therefore had none). However, the same sentence begins with the statement "Each landslide was assigned a location". Something can't be right. Please clarify.
R: The authors of the mentioned paper stated that they were able to retrieve the location of the landslides from the web information source only in 808 cases out of 1054. In their work, they carried on spatial analyses for all 1054 records and temporal analyses only for 808 cases. We thought it was worth mentioning this point.

To avoid misunderstandings, we reworded the sentence as follows: "Each landslide was assigned a location (possible only in 808 cases) and a date (daily resolution, in all cases)".

L125: Please check the work of Kirschbaum et al. (2010) again. If I remember correctly, they did not compile worldwide landslides of the years 2003 to 2008, but of the three years 2003, 2007 and 2008.
R: Thank you for pointing out this. You remember well! We corrected the text accordingly.

**Study area**

The section is concise and precise. I have only one small question/issue. According to Figure 2a, the vast majority of landslides occur in the Alps and the Apennines (these two mountain ranges are described in section 3.

Furthermore, the islands/regions of Sicily and Sardinia, as well as the region of Calabria, show numerous events.

(a) Would it be worth mentioning in line 143 that Sicily and Sardinia are hilly/mountainous?
R: We added the following sentence:" The latter also have a hilly/mountainous territory.".

(b) Do the hills and mountains of Calabria count as part of the "Appennino meridionale" or should something additional be mentioned here very briefly in the text?
R: Yes, the hills and mountains of Calabria count as part of the southern Apennines.

Figure 1: I suggest extending the caption a little. For example, how about:

"Study area: the Italian peninsula. Background from Bing; EPSG: 4326." Or:

"Study area: The peninsula of Italy, including the two main islands Sicily and Sardinia. Background from Bing; EPSG: 4326."
R: We modified the caption as follows: "Study area: the entire Italian territory. Background from © Microsoft; EPSG: 4326."

**Data and methods**

This is not absolutely necessary, but as section 4 is somewhat longer than all the others, subtitles might be helpful here. E.g. 4.1 Structure, 4.2 Information source, 4.3 Landslide type, 4.3 Spatial information, 4.4 Temporal information, 4.5 Organization and quality control.
R: We would prefer to keep the text without adding the subtitles.

L173-74: By "uneven" do the authors mean "inconsistent"? And right after that, by "conflicting" do they mean "contradictory"?
R: Yes, we changed the text using the suggested words.

L185: Consider rephrasing to "The catalogue contains the following information for each record"
R: We modified the text as suggested.

L187: "Date" is the combination of "day, month, year". It is OK to differentiate in Table 1; here in the parenthesis, however, the specification of "date" is rather confusing.
R: We changed dates to full dates. We prefer to specify it to reference all fields in Table 1.

Table 1: Consider expanding the caption slightly as follows: "Summary of fields included in the ITALICA catalogue."
R: We would keep only ITALICA since this acronym includes "catalogue"

L192: Since immediately above this text passage (in lines 185-187) an enumeration (i) to (iv) is already made, I think the enumeration (i) to (ii) can be omitted here. It is not really necessary.
R: Done, thanks.

L193: Consider changing to "news websites,"
R: Done, thanks.

Table 2: It is obvious, but perhaps it should still be pointed out in the caption that all the terms in Table 2 have been translated from Italian.

Are the original key terms (in Italian) available somewhere (e.g. supplementary material)? That

would be a nice-to-have, but is not absolutely necessary.

R: We added a specification: "Terms translated from Italian language.". We think it is not worth adding a supplementary material with Italian terms.

L214: While the adjective "reliable" fits well, the adjective "accurate" seems a bit strange to me here. "Accurate information about the exact or approximate landslide location", does that make sense? I would change it to: "...provide reliable information about the exact or approximate landslide location"

R: We changed the sentence as follows: "they provide reliable and accurate information about the landslide location in space and time…"

L220-22: I don't quite understand this sentence (or rather the last part of it). If the sources lack a precise description of the process, how can an event be assigned to one of these five types? Is the most likely process simply assumed for the given location? Please check sentence again.

R: Thanks for this comment. We meant that only the last class (generic shallow landslides) was assigned in the cases where the description of the type of landslide was missing in the information sources. We modified the sentence accordingly.

L236: I am not sure what the authors mean by "sighted"? Wouldn't "visible" be better? Is it meant here that a landslide was visible in Street View (if not, a new sentence would have to be started, instead of semicolon).

R: Yes, we meant that a landslide was visible in Google Street View. We modified the sentence accordingly.

L237: I suggest deleting the parenthesis "(i.e., 1 km2 or less)". This specification is clearly stated in line 230 and in Table 1 and is not necessary again here.

R: We modified the sentence accordingly.

L241: From line 232 to line 241 it is clearly explained when P0, P10 or P100 were attributed respectively.

According to Table 1, however, there is also P300. For the sake of completeness, perhaps P300 should also be mentioned briefly?

R: We added the following sentence: "Finally, a very low level of geographic accuracy $P_{300}$ was given in the few cases where only the municipality is mentioned."

L244: The text regarding T2 is a bit awkwardly worded; or more specifically, I stumbled across the verb "inferred". Would the following suggestion be correct in terms of content?

"T2 when the part of the day is known or the time can be estimated reasonably well."

R: We modified the sentence as follows: "$T_2$ when the part of the day is known or the occurrence time can be inferred"

**Description of the catalogue**

L266-67: This sentence can be misunderstood. The higher number of events results because the local authorities assisted in obtaining data, right? If necessary, rephrase, e.g.:

"Some areas have a higher concentration of events due to special agreements with local authorities for data collection."

R: Thanks for the comment. Actually, our agreements with local authorities were not only dedicated to data collection. Thus. we modified the sentence as follows "due to specific agreements with local authorities aimed at improving landslide forecasting."

L273: Change to "were recorded" or "information on more than 1000 landslides was collected"

R: We changed to "were recorded".

L283-84: Regarding the statement: "The difference between the mean and the median values is significant for all months,"

I disagree for the month of August, where the difference is very small. This should be noted here, for example in brackets.

R: Thanks, we added this note.

Figure 2: Consider the following two minor additions to the caption:

"Location of the catalogued landslides in ITALICA,"

"The number of landslides of each type is given in brackets in the legend."
R: We modified the text accordingly. Thanks.

Figure 3: Consider slightly rephrasing the caption as follows:

"Violin plot of the monthly distribution of landslides in ITALICA. Months are grouped into four seasons: Winter DJF (December–January–290 February), spring MAM (March–April–May), summer JJA (June–July–August), and autumn SON (September–October–November). The number of landslides in each season is shown in the legend."
R: We modified the text accordingly. Thanks.

Table 4: The data set covers 26 years. Strangely, however, the mean values of the individual months in table 4 were not calculated using the 26 years. For example: January has a total of 544 landslides; 544/26= 20.9; however, 23.67 is given as the "mean"; this would correspond to an investigation period of 23 years with 544 events.

None of the mean values of the months in Table 4 seem to have been calculated on the basis of 26 years (this varies between 19 years in March and 23 years in January and October.
R: Thanks for the comment. You are right, Table 4 showed incorrect values! We modified Table 4 by calculating the mean, median and landslide number in the time period 2002-2021 (which is the most representative period). We also moved Table 4 after Figure 4, which shows the scarcity of landslides in the first 6 years (1996-2001).

L296-97: I think this sentence does not quite describe Fig. 4 correctly. My suggestion:

"For each year, the mean and median values of the monthly number of landslides are given in the corresponding panel."
R: We modified the text accordingly.

Figure 4: The first six years of the study period show hardly any events. This observation is noted in lines 297-298 by the authors. In fact, the catalogue averages around 11 events per month for the first six years (1996-2001) and around 312 events per year for the following 20 years. That is a massive difference. Is this solely due to the weather (i.e. hardly any heavy precipitation events triggering landslides) or is there also a methodological reason (i.e. that the data collection in later years allowed the identification of more events). I think it would be good if the authors could briefly explain or discuss this difference somewhere in the text.
R: You are right. We now added (line 296): "The limited number of events does not depend on particular weather conditions during the period, but on the fact that historical data were available for only a few regions."

L300: Instead of "from 2007 to 2021" I suggest "between 2007 and 2021"
R: Done.

Also, it is not entirely clear which four years the authors mean; presumably 2007, 2017, 2020 and 2021. Median and mean values are also close to each other for 2015 and 2016 (with rather low intra-annual variability). Perhaps this could be made more precise.
R: Thank you for the comment. The sentence was not clear. We modified the text accordingly: "Except for the years 2013, 2014, and 2019, all others show similar mean and median values."

L321-23: I feel that this important information should be stated before the authors begin to describe the well-illustrated Figure 5. Consider moving this sentence before line 306.

Please also refer to Figures 5c and 5d in this sentence, as the exclusion of the 1996-2001 data only affects this figure (as far as I understand). E.g. "We excluded the period 1996-2001 from the analysis in Figures 5c and 5d…"

L328: It is not clear whether this statement ("Overall, more than a third of the catalogue (2175) is highly accurate...") refers to the entire catalogue or to the reduced catalogue (2002-2021). Please clarify.
R: Thanks, the statement refers to the entire catalogue. We clarified this in the text.

Figure 6 In the current version of the figure, the information content was always assigned to only one source (IR or NR). Presumably this is the "main source". However, in lines 215-217, the authors mention that the temporal and spatial accuracy could often be improved by combining institutional reports with news reports. Would there be a way to show this in Figure 6?
R: Unfortunately, it is not easy to trace that detailed information now. During the data collection phase, we did not find it to be a relevant information, sorry!

**Remarks and conclusions**

Consider renaming this section "Concluding remarks" because for a classic/typical chapter "Conclusions" it is a bit lengthy (compared to the length of the whole text).
R: Done, thanks.

L344-45:     Consider changing to the more straight forward

"…of rainfall-induced landslides with precise spatial and temporal localisation currently available…"
R: We accepted the suggestion.

L345-47:     In lines 345-347, I can understand the authors' reasoning. However, I find the two sentences not optimally formulated. I make an alternative wording suggestion below and let the authors decide how they want to proceed.:

"The selection criteria are relatively strict compared to other inventories. At the lowest acceptable geographic accuracy level, P300, the location of the landslide can still be indicated within a radius of less than 10 km. And the worst level of temporal accuracy, T3, still requires that at least the day of occurrence of the landslide is known, which is the usual maximum accuracy in most catalogues."
R: We modified the text as suggested. The new text reads: "The selection criteria are relatively strict compared to other inventories. In particular, at the lowest acceptable geographic accuracy level $P_{300}$ the location of the landslide can still be indicated within a radius of less than 10 km. Similarly, the worst level temporal accuracy $T_3$ still requires knowing at least the day of occurrence of the landslide which is the usual maximum accuracy in most catalogues."

L361ff:     The paragraph from lines 361 to 366 is still about spatial and temporal accuracy. In my opinion, it should therefore be attached directly to the first paragraph (after line 355). The statements regarding required manpower and time (now lines 356 to 360) can follow afterwards.
R: We prefer to keep this arrangement of the text because this part allows us to introduce the catalogue applications that we have made in the past - applications that you rightly suggested should be added in your initial comments and that we have now added in the manuscript.

I would also make the first sentence more concise: "The high spatial and temporal accuracy of ITALICA is its main strength."
R: Done.

L370ff:     I do not understand the reasoning of the authors in lines 370-374. Are they saying that the database is only temporally homogeneous (complete?) for those administrative regions with which collaboration agreements exist and not for all other regions?
R: Yes, that's what we meant.

The statement about "completeness" also made me wonder. Isn't it possible, for example, that in a very dry year in one region only very few landslides occur and thus only a few events can be recorded?
R: Right! We modified the text as follows "Excluding years recognized as extremely dry, a possible functional …".

**2) Technical details**

L32:        Consider "disasters" instead of "catastrophes"

L33:        Consider changing to "a variety of causes, including…"

L33:        Consider changing to "…and anthropogenic factors."
L42:        Change to "rainfall-induced landslides"

L43:        Consider changing to "over large areas"

L45:        Consider changing to "…between rainfall and landslide occurrence,"

L47-48:     I suggest changing to:
            "…predicting the occurrence of shallow landslides, where there is a direct correlation between
            rainfall and landslide initiation. However, rainfall thresholds are…"

L57:        Consider changing to "…homogenous catalogue containing accurate information…"

L62-63:     Consider changing to "… in the literature, which are briefly reviewed in this paper."

L69-70:     Consider changing to:
            "…as part of the national project AVI – *Aree Vulnerate Italiane* (an acronym for Areas Affected by
            Landslides or Floods, Guzzetti et al., 1994). Subsequently, …"

L71:        "Guzzetti (2000) compiled…" instead of "(Guzzetti, 2000) compiled…"

L72:        "expanded by Salvati et al. (2010, 2018)." instead of "expanded by (Salvati et al., 2010, 2018)."

L79:        Consider changing to "Hervás (2012) published a detailed analysis of the (then) existing national
            landslide databases in Europe."

L90:        Rather than "triggering events", I would prefer "triggering factors" or "event triggering" in this
            context.

L130:       Here and in lines 266, 271 and 365 I would not capitalize the word "country"

L137:       Consider changing to "…covering 301336 km$^2$ in …"

L138:       Consider writing "mountain ranges" instead of "mountain chains"

L139:       Consider changing to "…west-to-east arc across the northern tip…"

L139-40:    Change to "…from east to west reaching an altitude of over 4,800 m a.s.l. and separating the Italian
            peninsula…"

L140:       Change to "The Apennines are a…"

L141:       Change to "Elsewhere, Italy plunges into the Mediterranean Sea and is surrounded in particular by
            the Adriatic,"

L147:       Consider changing to: "as it lies at the boundary between the Eurasian and African plates."

L150:       Change zo "In the north…"

L151:       Change to "…increasing towards the south."

L151-52:    Change to "…and prealpine areas as well as the northern Apennines have higher precipitation, with
            mean annual values exceeding 2,000 mm."

L155:       Consider removing the paragraph mark (avoid single-sentence paragraphs)

L162:       Consider changing to "…occurred in uninhabited areas or for which there are no institutional or
            news reports are rarely included in the ITALICA catalogue"

L164:       Change to "In order to obtain detailed and up-to-date…"

L168:       Change to "…in archives at different levels of…"

| | |
|---|---|
| L168: | Consider using "(municipal, provincial and national)" to avoid confusion with the archives of the US states |
| L170: | Change to "…involvement of institutions…" |
| L179: | Consider changing to "…such as the general location and the date of occurrence…" |
| L194: | Change to "…systematic searches of regional…" |
| L205: | Please add comma after "social media posts" |
| L210: | Use full stop instead of comma: "…and State Forestry Corps." |
| L210: | Instead of "were provided" I would use "are provided by ANAS". Or does that statement only apply to the past? |
| L214: | Consider changing to "…have proved to be particularly useful," |
| L224: | Change to "Landslides are then represented as points…" |
| L228: | Consider changing to "which make it possible to consult all the maps…" |
| L234: | Change to "with the exact kilometric indication" |
| L236: | Consider changing to "in a few cases," |
| L249: | Consider changing to "…to determine the inferred time of failure," |
| L250: | Consider changing to "Lastly, if the news only reports the day…" |
| L251: | Change to "… a daily temporal accuracy of T3 and is conventionally assumed to have occurred at the end of the day (23:59)." |
| L260: | Consider removing the paragraph mark (avoid single-sentence paragraphs) |
| L264: | Change to "…that occurred on the Italian territory…" |
| L275: | According to the text in line 270, the abbreviation for "mud flow" is MF (and not MD). Please clarify |
| L283: | Change to "…by a factor of 2.5," |
| L287: | Consider changing to "…are mainly located in the…" |
| L307: | Consider extending to: "…and two 10-year sub-periods 2002-2011 and 2012-2021." |
| L310: | Change to "…have a low and very low…" |
| L316: | In the figure caption, change text to "…in the overall observation period 1996-2021." |
| L318: | In the figure caption, change to "Refer to Table 1 for…" |
| L323: | Here and in line 325, replace "period" with the more accurate "sub-period" (introduced in line 322) |
| L324: | Change to "…has more than doubled from the sub-period 2002-2011 to the sub-period 2012-2021." or shorter "…has more than doubled from 2002-2011 to 2012-2021." |
| L327: | Consider changing to "…of the total events (Fig. 5d)." |
| L329: | Consider extending to "…and only daily (T3) temporal resolution." |
| L339-40: | In the figure caption, consider changing text to "…number of landslides (1996-2021) for which information was obtained from institutional (IR) or news (NR) reports, divided into classes of…" |
| L348: | Change to "…significantly limit the number…" |
| L349: | Consider changing to "As a result, an average of 40%..." |
| L350: | Use "produced" instead of "prepared" |
| L353: | Consider using "country" instead of "territory" |
| L353: | "policies" |
| L354: | Consider "is collected" instead of "is gathered" |
| L368-69: | Consider changing to "The main reason for this is the regional scale at which…" |
| L371: | The percentage refers to all entries in ITALICA; this would need to be specified, e.g. "(27% of all entries)". |
| L381: | Consider adapting to "it contains exclusively landslides induced by rainfall, unlike datasets presented in Guzzetti et al. (1994), Innocenzi et al. (2017), Calvello and Pecoraro (2018), which |

contain information on landslides…"

L387:    Change to "…a comparison with the Polaris…"

L392:    Consider changing to "…available in the literature report a small number of landslides on the Italian territory,"

L393:    Change to "…according to Kirschbaum et al. (2015); 72 landslides in the period 2005–2014 according to Haque et al. (2016); and 39 in the period 2004–2016 according to Froude and Petley (2018)."

L398:    Change to "to report landslides in their area, providing continuous feedback from the real world"

L400:    Change to "…suggesting its possible future…"

L404-05: Consider changing to "…and climatic contexts in Italy and can easily be used to compile new catalogues of high spatial and temporal accuracy in other countries."

R: we have accepted all the suggested technical corrections. Thank you very much for your careful review!

Table 3:   Why is the inferred time for the "Night" time slot not 04:00, which is practically halfway between 23:59 and 08:00?

At least in summer, 05:00 is almost the time of dawn.

R: This was a conservative choice so as to be sure to include all landslides that occurred in the night. In the few cases where the source mentioned landslides occurring at dawn, we searched for the time of sunrise on the day indicated.

---

## Author Comment (AC2)

**Review of manuscript ESSD-2023-61: "ITALICA, an extensive and accurate spatio-temporal catalogue of rainfall-induced landslides in Italy" By: Silvia Peruccacci, Stefano Luigi Gariano, Massimo Melillo, Monica Solimano, Fausto Guzzetti, Maria Teresa Brunetti**

Dear Editor and Authors.

Thank you for your valuable and detailed presentation and examination of the ITALICA dataset. I feel this is a good paper that clearly presents the difficulties, resources, and data variability available in compiling such an inventory. The focus of the work is clearly defined, and the steps/methodologies adhered to in order to compile such a dataset are clearly established.

I found the paper to be engaging and comprehensive, with questions arising mostly being answered as the paper progressed. The Figures and Tables all support the text and are positioned well to support the narrative.

I have a few very minor comments that I would like to see addressed to support the paper, most of which appear to fit with the Background section. Please see comments supplied.

I would recommend this paper be accepted for publication with very minor edits.
We really appreciate the positive comments and thank you for the helpful suggestions.
* * *
**Section 2: Background**

**Lines 66 – 133** Whilst presenting a good background of inventories, could this section be restructured slightly either as a list or by geographic area. E.g.  Italian databases are mentioned at the beginning and the end, Nicaragua in between two Italian databases. May I suggest reviewing this section and formatting along the lines of *Italy > Europe> rest of the world example > Global*.
R: Thanks for the comment. Originally, the background section was organized by listing chronologically the examples of landslide catalogues, inventories, and databases found worldwide. We acknowledge that this organization could be a bit confusing, therefore we agree on the effectiveness of presenting the review following a geographical order. We decided now to start form the global scale, followed by non-European cases, European examples, and finally Italian references. In this way we link the review of the Italian examples with the statements that stress the advantages of ITALICA. The new text of the section is now completely re-arranged.

- I would suggest that the authors review if they are stating examples or offering comparisons of contents.
  R: If we understood the Reviewer request, we stated at line 60: "… a brief review of which is given in this work". In our intention, the brief review reports the main information available from the cited literature. This may possibly imply an indirect comparison of the data by the reader.
- The text should clearly define the time limited databases and those which are still being updated. This is not always well presented.
  We did not report this information since it was not available.
    - L84       Change "included" to "to date includes" – the GB database is being updated constantly and is not a static inventory. It also is different to some of the other databases presented as it includes historic (preglacial) undated landslide event deposits not just present day (hence 17000).
      R: thanks for highlighting this. We modified the text accordingly.

    - L85       Poland details until April 2014 but from when – is this time stamped.
      R: We added the starting date of the mapping project (found in the related reference). We modified the text changing "until April 2014" into "from 2008 to April 2014". Please note that this is period during which the landslides were mapped, not the time window of landslide occurrence.

    - L94       Slovenia – ongoing but from when.
      R: We added this detail in the text.

- Line 84 suggest an edit "with capability to include" – data is only added where it is available although the fields are present as stated in references.
  R: Suggestion accepted, thanks.

**Section 4: Data and Models**

**Table 1** Geographic accuracy, the text is described "as geographic accuracy based on the area which the landslides realistically occurred described as radius from the coordinates" however the examples are presented as area. The approximate radius is mentioned later in the text (L235-239) but it would be helpful to the reader if the radius could also be included in the table along with the area to assist interpretation.
R: Thanks, we added it.

**Section 7: Remarks and conclusions**

**L375**    I would suggest the authors consider L375-377 "the catalogue…" should be reordered to the end of the paper as the introduction line to the beginning of the summary for the future of the inventory and follow on work and included in the final paragraph beginning L394.
R: We prefer to keep the phrase where it is, as it refers to the possibility of increasing the spatial homogeneity of the catalogue.

**Technical notes:**

There are many areas where there are single sentence paragraphs. I strongly suggest these are reviewed and removed.
R: Done.

L30       "catastrophes" change to disasters
R: Done.

L148      "along the length of the peninsula"
R: We prefer to keep the text as it is.

L160      "non- anthropized areas" suggest "areas not subjected to any anthropogenic change".
R: We changed it in "uninhabited areas".

L394      Natural disasters – This can be a controversial term in some current natural hazard/ social science circles. If authors agree I would strongly suggest change to "natural hazards". This is supported in the contact of the sentence.
R: We prefer to keep this term, since it is the term used in the NASA's "Landslide Reporter" project description (https://gpm.nasa.gov/landslides/guides/LandslideReporter_Intro_English.pdf).

L308      Colloquial "on the other hand" should be removed.
R: We substituted it with "In contrast".

L332      Colloquial "on the other hand" should be removed.
R: We substituted it with "About".

L273      Caption for Figure 2 mud flow abbreviation is presented as MD – presented as MF in table 1 and L219 and L268. Please check.
R: Done.

L385      Colloquial "on the other hand" should be removed.
R: We removed it.

---

## Author Comment (AC3)

Dear Editor, dear Authors

The manuscript by Peruccacci et al. highlights how essential reliable landslide datasets are in the elaboration of landslide thresholds, to be used in the development of landslide early warning systems. The authors also emphasize how landslide inventories are important tools in the hazard characterization and assessment and, they provide a very useful and quite up-to- date overview of existing landslide inventories around the world.

Despite some few examples, most of the landslide inventories have been prepared after the year 2000 and were created with the purpose to show where historical landslide events have occurred and the damages they have caused. In the compilation of these inventories, the main concern was mainly the quantity of data rather than the quality, or information about triggering mechanisms. Most of these inventories were prepared as part of specific projects and created during the duration of the project, not always updated, with new entries or improved in their quality, in later years.

The authors highlight quite well the difficulties in the compilation of landslide inventories and the fact that this is a time-consuming task. And this is, in my opinion, the reason why the compilation of landslide inventories is often a forgotten task in the national framework for landslide risk assessment, with very limited resources and personnel assigned, well indicated by Reviewer 1.

We must also remember that in the past many previous authors have prepared landslide thresholds by using separated landslide dataset. It is with the most recent need of operational landslide early warning systems that we demand reliable landslide data to use in thresholds analyses.

The authors described well the existence of other databases in Italy, their limitations and the need for a "separate" dataset for rainfall-induced landslides. This may seem a good solution at the beginning, but in a long run it will create problems for those who manage landslide prevention at national level. Therefore I have some general comments for the authors that I would like to be adressed in the document.

We thank the reviewer for the positive comments and helpful suggestions that significantly improved our paper.

- It is not clear if and how ITALICA communicate with the other existing inventories. If a new rainfall-induced landslide should be registered in ITALICA, will this be send automatically in the other databases and viceversa? How to deal with future updates? Who is responsible for these updates?

  R: ITALICA does not communicate with other Italian landslide inventories because it has different characteristics and purposes, as stated in the text (from line 378 et seq.). As explained in the Introduction, it has been more than 15 years that properly trained CNR IRPI staff have been involved in data collection (lines 151-152). The decision to start a new catalog in 2007 was necessary because of the lack of accurate information in the then existing Italian catalogs (AVI and IFFI). ITALICA is continuously being updated and new releases will be released periodically on Zenodo under our responsibility (lines 375-377). To be more clear, we added that the catalogue will be updated "by us".

- Have you considered the possibility to improve one of the existing databases instead of making a new one? Adding for example a "quality level code" or create a separate module with specific parameters for those landslides that should be used for thresholds?

  R: The possibility of improving existing databases was not feasible because the information needed to define a quality level code was not available or could no longer be found at the time of data collection. We added as a final remark in the Concluding remarks the following sentence: "ITALICA can certainly serve as an example for the collection of new accurate data for setting rainfall threshold.".

- Some operational early warning systems send alert messages also for landslide triggered by snow melt or only for high soil water saturation, what do you suggest based on your experience? do we need to create a separate database for each triggering?

  R: Currently, landslides triggered by the combined effects of precipitation and snowmelt have not been included in ITALICA (as stated in line 180). Using this type of data would involve using reliable algorithms and temperature data to derive the water equivalent of snow in mountainous areas. At present, we have not invested time and resources in this topic. In our opinion, it would be better to identify snowmelt triggered events in databases with a flag.

- Many countries have one database, managed by one national institution, with limited resources and personal, and efforts are being made to improve the quality of the registered events, by assigning a quality level to each entry and to provide information about triggering. Based on your experience and take into account the problems with future updates, what do you recommend? To create separate databases or improve the existing inventories?

  R: There is no nationally managed database in Italy that simultaneously contains information on landslide cause, time of initiation, and location with sufficient levels of accuracy as in ITALICA. For our

purposes, it was easier to start a new collection than to search for missing information in existing databases, which were not designed to predict the spatial and temporal occurrence of shallow landslides (line 41et sec.).

- At the end you have created a database with both low and high quality data. Could you explain better if you could use the low quality data in your threshold analyses (maybe for threshold at regional scale requiring low accuracy)? or do you need always high quality data?

  Thank you for the comment. In ITALICA the classification of data by quality (from very low to very high) refers only to spatial accuracy. In particular, the percentage of data with low ($P_{100}$) and very low ($P_{300}$) spatial accuracy is only 4.5% (lines 307-308). When calculating rainfall thresholds, it is suggested that these data be excluded. In this regard, we added the following sentence after line 361: "For this application, we suggest excluding records with low ($P_{100}$) and very low ($P_{300}$) geographic accuracy".

I also agree that the article is quite well structured and written, with adequate length and, with explanations and statements well illustrated.

It can be an important contribution and of interest to the landslide research community
working in particular with landslide early warning systems. I would recommend this paper to be accepted for publication, with very minor edits (see supplied comments).

I have just few comments/edits on the text

- Page 2 line 46, "shallow landslides" maybe useful to add what kind of types are in this category

  R: We added "(e.g. slides, flows, and falls)".

- Background: In the list of inventories, I have not seen this reference. *Herrera, G., Mateos, R.M., García-Davalillo, J.C. et al. Landslide databases in the Geological Surveys of Europe. Landslides **15**, 359–379 (2018). https://doi.org/10.1007/s10346-017-0902-z* . Please check if they have mentioned other inventories that are not in your review.

  R: Thank you! We added the suggested reference. The text now reads: "Herrera et al. (2018) provided an update of the previous European recognition, collecting 20 national landslide databases including 849,543 landslides of different types (528,903 in Italy)."

- Another reference that is missing is the one that describe the inventory for Norway: Jaedicke, C., Lied, K., and Kronholm, K.: Integrated database for rapid mass movements in Norway, Nat. Hazards Earth Syst. Sci., 9, 469–479, https://doi.org/10.5194/nhess-9-469-2009, 2009.

  R: Thank you! We added the suggested reference. The text now reads: "In Norway, Jaedicke et al. (2009) collected a database of more than 33,000 rapid mass movements of different types, e.g. including also snow avalanches and subaqueous slides, without detailed temporal information (only the year of occurrence is often now)."